

# Age, thinning and spatial origin of the Beyond EPICA ice from a 2.5D ice flow model

Ailsa Chung[1], Frédéric Parrenin[1], Robert Mulvaney[2], Luca Vittuari[3], Massimo Frezzotti[4], Antonio Zanutta[3], David A. Lilien[5], Marie Cavitte[6], and Olaf Eisen[7,8]

[1]Université Grenoble Alpes, CNRS, INRAE, IRD, Grenoble INP, IGE, 38000 Grenoble, France
[2]British Antarctic Survey, Cambridge, UK
[3]Department of Civil, Chemical, Environmental and Materials Engineering, University of Bologna, Bologna, Italy
[4]Department of Science, University Roma Tre, Rome, Italy
[5]Department of Earth and Atmospheric Sciences, Indiana University, Bloomington, IN, USA
[6]Earth and Life Institute (ELI), Université catholique de Louvain-La-Neuve (UCLouvain), Louvain-la-Neuve, Belgium
[7]Glaciology, Alfred-Wegener-Institut Helmholz-Zentrum für Polar-und Meeresforschung, Bremerhaven, Germany
[8]Fachbereich Geowissenschaften, Universität Bremen, Bremen, Germany

**Correspondence:** Ailsa Chung (ailsa.chung@univ-grenoble-alpes.fr), Frédéric Parrenin
(frederic.parrenin@univ-grenoble-alpes.fr)

**Abstract.**

The European Beyond EPICA – Oldest Ice consortium is currently conducting an ice core drilling project at Little Dome C (LDC) in Antarctica with the aim of retrieving a continuous ice core up to 1.5 Ma. In order to determine the age of the ice at a given depth, 1D numerical models are often employed. However, they do not take into account any effects due to horizontal

flow. We present a 2.5D inverse model that determines the age–depth profile along a flow line from Dome C (DC) to LDC that is assumed to be stable in time. The model is constrained by dated radar internal reflecting horizons. Surface velocity measurements are used to determine the flow line and ascertain the flow tube width, which also allows the model to consider lateral divergence. This new model therefore improves on the results produced by 1D models previously applied to the DC area. By inferring a mechanical ice thickness, the model predicts either the thickness of a basal layer of stagnant ice or a basal

melt rate.

Results show that the deepest ice at Beyond EPICA Little Dome C (BELDC) originates from around 15 km upstream. The oldest ice with useful age resolution, i.e. with an age density of 20 kyr m$^{-1}$, is predicted to be 1.12 Ma at BELDC. Over the LDC area, the 2.5D model predicts a basal layer 200–250 m thick at the base of the ice sheet. Modelled ice particle trajectories suggest that this layer could be composed of stagnant ice, accreted ice or even disturbed ice containing debris. We explore the

possibilities, though this is an open question that may only be answered by analysis the Beyond EPICA ice core once it has been drilled. Finally, we discuss in detail a thinning in the basal layer which is less than predicted by the model, as observed in other ice cores. This could mean that modelled ages are significantly over-estimated in the deepest part of the ice column. Given that the age estimate from the 2.5D model is younger than previous estimates, we suggest that horizontal flow is an important factor in this region. However, our model assumes that the flow line features such as flow direction and dome location have not

change over the time period considered, which might not be the case.





# 1 Introduction

The Beyond EPICA – Oldest Ice project aims to retrieve a continuous ice core record covering the past 1.5 Ma from Little Dome C (LDC)—a secondary dome ∼35 km south-east of Dome C (DC) in Antarctica (Parrenin et al., 2017; Chung et al., 2023a). Age–depth models are invaluable for both selecting promising ice core drill sites and providing age constraints once the ice has been extracted. This will be especially important for the deepest ice at the Beyond EPICA drill site (BELDC), as the age density (number of years per depth unit) is likely to be very high (Chung et al., 2023a), making extracting a paleoclimatic signal challenging. Using radar internal reflecting horizons (IRHs) to constrain age–depth models is a well-established method that has been applied to many regions (Waddington et al., 2007; Koutnik, 2009; Steen-Larsen et al., 2010; Parrenin et al., 2017; Lilien et al., 2021; Obase et al., 2023; Chung et al., 2023a; Wang et al., 2023). By looking at the depth of IRHs at an ice core site, we can use existing ice core chronologies to attribute ages to IRHs, giving us isochronal layers throughout the ice sheet (e.g. Cavitte et al., 2020). Models can then compare results to the observed isochrones to assess how well the model has performed. Or isochrones can be used directly as constraints, with models fitting to these "tie points" on the age–depth profile.

A 1D inverse model constrained by radar isochrones was applied in the regions of DC (Lilien et al., 2021; Chung et al., 2023a) and Dome Fuji (Wang et al., 2023) using an inferred mechanical ice thickness to determine a basal melt rate or the thickness of a layer of stagnant ice. At LDC, this basal unit has been observed in radar surveys by Cavitte (2017) and Lilien et al. (2021). Further evidence for a stagnant ice layer was presented by Chung et al. (2023a), who showed that the basal unit seen in radar surveys is of comparable thickness to a stagnant ice layer predicted by a 1D inverse age–depth model. They then corroborated this with vertical velocity measurements made with an autonomous phase-sensitive radio-echo sounder (ApRES). There is a general consensus that the basal layer is around 200–250 m thick at the BELDC drill site. Possible origins of this basal layer include shear between the top of the stagnant ice layer and the dynamic ice above, refrozen ice (Bell et al., 2011), an anisotropic crystal fabric which is resistant to compression or a layer with considerable distortions on the micro scale (Drews et al., 2009).

1D models are valuable tools as they are simple and can be run relatively quickly. However, they do not account for horizontal advection which can be an important factor near a dome (Koutnik et al., 2016) as it can affect the interpretation of an ice core's age–depth profile downstream (Fudge et al., 2020; Gerber et al., 2021). A 2.5D model considers vertical and horizontal velocity along the flow line, with a finite width of the flow tube perpendicular to the flow line. 2.5D inverse models that have been developed thus far have used a Monte Carlo method to fit to isochrones (i.e. random sampling rather than intentional minimization of misfit) and required a steady-state assumption (Steen-Larsen et al., 2010). A steady-state model assumes parameters such as ice thickness and accumulation rate have remained constant over the considered time period. Waddington et al. (2007) and Steen-Larsen et al. (2010) applied a steady state 2.5D model to Taylor Mouth in Antarctica to determine past accumulation rates. Transient models can account for temporal changes but are computationally demanding due to the increased number of parameters allowed to vary (Koutnik and Waddington, 2012). The pseudo-steady assumption has the advantage of a steady vertical velocity profile and geometry but accumulation rates which vary over time (Parrenin et al., 2017). This offers



some middle ground as it better represents real conditions than the steady state assumption but has lower computation time
than a transient model.

In this study we use a 2.5D pseudo-steady model to assess the suitability of the 1D assumption at DC (i.e., that horizontal
flow is negligible). Applying a 2.5D inverse model also allows us to investigate the trajectories of ice particles and therefore the
location of their deposition upstream. This is of particular interest at the BELDC drill site because even slow flow can cause
substantial horizontal displacement on the million-year timescales targeted by the core. The aim of this study is to investigate
the impact of horizontal advection on our age predictions along the DC–LDC flow line and discuss the model's applicability
to other regions. In Sect. 2, we describe the 2.5D inverse model used in this study. We also give details on constraining
observations like surface velocity measurements at LDC, used to determine the flow tube, and the radar IRHs. In Sect. 3, we
present the ice particle trajectories, modelled ice age and implications for the BELDC ice core. In Sect. 4, we compare the
model results to previous studies and discuss the advantages and limitations of our approach.

## 2 Methods

The forward model is based on the analytical development presented in Parrenin and Hindmarsh (2007) with a coordinate
transformation to a system where particle trajectories are linear and therefore straightforward to calculate. There is no direct
thermal representation in the forward model as we use an inferred mechanical ice thickness to determine a basal melt rate. For
the inverse model, as in the 1D approach (Chung et al., 2023a), there are three inferred parameters—the steady-state (i.e., time-
independent) accumulation rate $\bar{a}$, the Lliboutry velocity profile parameter $p$ and the mechanical ice thickness $H_m$. The flow
tube width was determined using geodetic surface velocity measurements (Sect. 2.3) and the inverse model is constrained by
isochrones along a radar transect which approximately follows the flow line (Sect. 2.4). The forward model is run for different
values of the three inferred parameters, resulting in modelled ages for observed isochrones. The inverse model optimises a cost
function by selecting the best-fit parameters, which minimise the age misfit of the isochrones generated by the forward model
to the observed isochrones.

### 2.1 Forward model

We use a pseudo-steady model which includes a steady-state geometry and velocity profile. A multiplicative temporal variation
factor is applied to the steady accumulation rate ($\bar{a}$, Parrenin et al., 2017; Chung et al., 2023a). The accumulation variations
are those inferred from the AICC2023 chronology for EDC (Bouchet et al., 2023). The model is 2.5D meaning it accounts
for vertical and horizontal flow, as well as divergence along the flow line. The horizontal flux shape function is defined by the
Lliboutry vertical velocity profile ($\omega$, Lliboutry, 1979), which depends on the $p$ parameter,

$$\omega\left(\zeta\right) = 1 - \frac{p+2}{p+1}\left(1-\zeta\right) + \frac{1}{p+1}\left(1-\zeta\right)^{p+2}, \tag{1}$$

where $\zeta$ is the vertical coordinate normalised to 0 at the bedrock (mechanical ice thickness) and 1 at the surface. There are
several assumptions associated with the Lliboutry profile including that the ice is isotropic, bedrock variations are smooth and





the horizontal velocity is non zero at the modelled location. However, changing the value of $p$, making the vertical velocity profile more or less linear, can approximate the effect of relaxing these assumptions.

We use the equation for ice particle trajectories presented in Parrenin and Hindmarsh (2007) and build a grid that follows these trajectories. This method has the advantage that ice particle trajectories pass exactly through grid nodes so no interpolation is required in the forward model. Given the increasing horizontal flow speed along the flow line, this means that the grid along

the $x$ axis is finer near the dome and coarser further downstream. In the $z$ direction, the grid is coarse near the surface and becomes finer towards the bed where ice layers have thinned considerably (Fig. S1).

A key difference from Parrenin and Hindmarsh (2007) is that the forward model has no basal melting parameter; it instead uses an inferred mechanical ice thickness from which a basal melt rate or the stagnant ice thickness can be determined (Sect. 2.2 and Chung et al., 2023a). A full description of the forward model will be available in a separate subsequent article.

## 2.2 Inverse model

We infer three parameters at each horizontal position, $x$: steady accumulation, $\bar{a}$ (as defined in Sect. 2.1); thinning parameter, $p$ (Eq. 1); and mechanical ice thickness, $H_m$. We perform the inversion at each point along the flow line simultaneously. In order to prevent $a < 0$, $H < 0$ and $p < -1$, we use variables $a = e^{a'}$ $p = e^{p'} - 1$ and $H_m = e^{H'_m}$. To optimise the model, we minimise the squared deviation of modelled ages and input parameters from expectations,

$$S = \sum \frac{\left(\chi_{obs}^i - \chi_{mod}\left(d_{obs}^i\right)\right)^2}{\left(\sigma_{obs}^i\right)^2} + \frac{\left(p'_{prior} - p'\right)^2}{\left(\sigma^{p'}\right)^2} + \frac{\left(a'_{prior} - a'\right)^2}{\left(\sigma^{a'}\right)^2} + \frac{\left(H'_{obs} - H'_m\right)^2}{\left(\sigma^{H'_m}\right)^2},$$ (2)

where $d_{obs}^i$, $\chi_{obs}^i$ and $\sigma_{obs}^i$ are the depths, ages and age uncertainties of the $i^{th}$ observed isochrone, respectively. $\chi_{mod}$ is the modelled age. We use $p_{prior} = 3$ and $a_{prior} = 0.02$ m yr$^{-1}$, similar values to EDC, and $H_{obs}$ is the ice thickness observed in the radargram. The confidence intervals $\sigma^{a'}$, $\sigma^{p'}$ and $\sigma^{H'_m}$ are set to 1 to allow the function to vary within reasonable limits. We use the Python Scipy least squares optimisation with the Trust Region Reflective algorithm to find the parameters that best

minimise $S$ (Eq. 2). The inverse model uses fewer horizontal grid points than the forward model which reduces computation time without significant compromise to the final fit achieved.

The difference between the mechanical ice thickness, $H_m$ and $H_{obs}$ is used to determine either a basal melt rate $m$ or the thickness of a stagnant ice layer, as in Chung et al. (2023a). Where $H_m < H_{obs}$, there is a stagnant ice layer of thickness $H_{obs} - H_m$. Where $H_m > H_{obs}$, there is basal melting $m$ which is calculated using the value of the ice flux $\Delta q$ at $H_{obs}$,

$$m = \frac{\Delta q(H_{obs})}{Y \Delta x},$$ (3)

where $Y$ is the flow tube width.

### 2.3 GNSS surface velocity data

We determined horizontal ice flow in the DC–LDC region using an existing network of survey poles, supplemented by 14 additional poles (Figure 1). The existing network of 38 poles was installed in 1995/96 in the framework of the EPICA project



(black arrows in Fig 1, Vittuari et al., 2004). Using the Global Navigation Satellite Systems (GNSS) processed relative to a continuously operating base station (known as DCRU) located at Concordia station, geodetic measurements were made annually from 2005-2019. The 14 additional poles placed around LDC were surveyed using GNSS to determine ice-flow velocities, (blue arrows in Fig. 1).

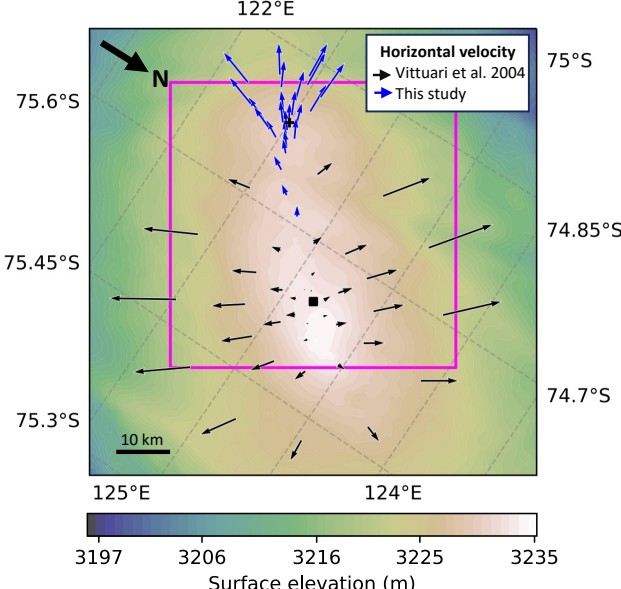

**Figure 1.** Contours show surface elevation. In blue are the new horizontal velocities at LDC. Black shows velocities from Vittuari et al. (2004). BELDC is marked by a black cross. EDC is marked by a black square (note: this is also the location of DCRU). The pink box highlights the area shown in Fig. 2.

The surface ice velocities observed at the top of DC are extremely small (a few cm per year), which is the same order of
magnitude as motion due to plate tectonics. The observed displacements are the result of bedrock motion (due to continental drift) and ice flow due to deformation under the force of gravity. The analysis was carried out using the Bernese GNSS Software developed at the University of Bern which uses a classical differentiated approach. Given the small magnitude of the expected ice speeds, it was necessary to guarantee high repeatability in centring the geodetic antennas during the stationing phases at the different sites. For this reason, aluminium poles were used with a forced centring system of the antennas in the upper part
of the pole.

In order to avoid misinterpreting tectonic plate movement as ice flow, the slow velocities necessitate considering ice motion relative to the top of DC. The absolute velocity (due to plate tectonics and ice dynamics) of DCRU over the period 2005-2019 was 10.4±0.4 mm/yr at 275° relative to true north and the GNSS baselines at LDC were calculated using the position of DCRU as the reference. The poles at LDC were initially measured by GNSS for 2–3 days in December, 2016, then resurveyed for
2–3 days in December, 2017. From the comparison of the positions of the poles obtained in subsequent periods, the ice surface





velocities were estimated, Table S1. The predicted motion of bedrock in the LDC region was obtained from the International Terrestrial Reference Frame 2014 Eulerian pole for the Antarctic plate, which gives an almost constant value of 11.5mm/yr at 180° relative to true north.

In order to account for ice flow divergence, the forward model requires a flow tube width which we determined using surface
velocity data, derived by GNSS around DC and LDC. Using the Scipy grid data function, 2D interpolation of the velocity over the DC region was done. The direction of flow was determined by selecting a point downstream and backtracking i.e. following the direction of the surface velocity upstream to the dome. This backtracking process was repeated for two downstream points adjacent to the central flow line, the distance between the two resulting flow lines gave the flow tube width (grey shaded area in Fig. 2). The specific central flow line we look at was selected as it passes through both the EDC and the BELDC drill sites.
This allows us to date the IRHs using the EDC chronology (Bouchet et al., 2023), and model the age scale for BELDC.

## 2.4 Radar

In order to constrain our model, we require traced radar IRHs along the flow line from DC to LDC. There are currently no radar transects which follow the DC–LDC flow line exactly, so we map IRHs which pass close by (red line in Fig. 2). The radar transect begins at EDC whereas the flow line begins at the summit of DC, therefore we refer to the EDC–LDC radar
transect and the DC–LDC flow line. We use 19 IRHs along the radar transect from EDC to LDC which were traced by Chung et al. (2023a) and dated between 73 and 476 ka. Using the newest chronology for the EDC ice core AICC2023 (Bouchet et al., 2023), we have dated these IRHs again (Table 1), which explains why ages might differ from previous studies. Employing the newest chronology reduced the age uncertainties of shallower isochrones, though absolute ages only changed up to a maximum of ±2.3 ka compared to previous studies with AICC2012 (Bazin et al., 2013). We map the radar IRHs onto the DC–LDC flow
line by taking the point of intersection between the radar transect and a line perpendicular to the local direction of flow. This allows us to run the model along the flow line with IRH constraints.

## 3 Results

Here we present the results when the 2.5D model was applied to the DC–LDC flow line. We show the ice particle trajectories, predictions for the basal melt rate or stagnant ice thickness as well as the age of the deepest well-stratified ice.

### 3.1 DC–LDC flow line

The central flow line was determined by following the surface velocity direction upstream (Figure 2), backtracking from LDC towards DC (Sect. 2.3). The same process was then followed for two points 5 km either side of LDC, as this allows us to take into account the divergence along the ridge without incorporating too much flank flow on either side. The distance between the two offset flow lines gave the flow tube width, which was then normalised to its widest point (Figure 2). At the point where the
flow tube width decreased to below 10 m, the process was restarted. This is because when the flow tube becomes too narrow, variations in the flow tube width drop below measurement precision. This method brought the flow tube to around 2 km from





| Isochrone | TWT at EDC | Depth | Age | Age uncertainty |
|---|---|---|---|---|
| | (ns) | (m) | (ka) | (±ka) |
| IRH_1 | 12689 | 1079 | 73.2 | 1.5 |
| IRH_2 | 14192 | 1206 | 84.8 | 1.5 |
| IRH_3 | 14958 | 1270 | 90.8 | 1.7 |
| IRH_4 | 15808 | 1342 | 97.7 | 1.7 |
| IRH_5 | 17772 | 1507 | 115.8 | 1.6 |
| IRH_6 | 18827 | 1596 | 122.9 | 1.2 |
| IRH_7 | 20616 | 1747 | 133.2 | 1.6 |
| IRH_8 | 22303 | 1889 | 159.6 | 3.1 |
| IRH_9 | 23349 | 1977 | 178.2 | 3.6 |
| IRH_10 | 24748 | 2095 | 202.5 | 2.8 |
| IRH_11 | 25575 | 2165 | 214.4 | 2.8 |
| IRH_12 | 26873 | 2274 | 239.9 | 2.9 |
| IRH_13 | 27135 | 2296 | 243.2 | 2.5 |
| IRH_14 | 29380 | 2486 | 305.3 | 6 |
| IRH_15 | 29851 | 2525 | 320.4 | 5.8 |
| IRH_16 | 30555 | 2584 | 335.7 | 4.5 |
| IRH_17 | 31287 | 2646 | 367.3 | 10.6 |
| IRH_18 | 32004 | 2706 | 399.5 | 7.6 |
| IRH_19 | 33417 | 2826 | 475.3 | 17.9 |
| Bedrock | 38325 | 3239 | - | - |

**Table 1.** The ages of the radar IRHs based on the AICC2023 chronology (Bouchet et al., 2023). IRH two-way travel times (TWTs) and depths are taken from Chung et al. (2023b).

EDC, at which point the surface velocity is too low as we approach the dome and the flow tube width tends to zero too quickly. Therefore, the flow tube width was then exponentially extrapolated to the summit of the dome. The DC–LDC flow line is defined as shown in Fig. 2 and we refer to this flow line in the subsequent figures. The EDC–LDC radar transect does not

follow the exact flow line, so the isochrones were mapped onto the flow line (Sect. 2.4). Figure 2 shows the flow line starting at the high point of DC where flow tube width is near zero, then increases along the ridge and towards LDC. In all subsequent subsections, we consider the flow line with mapped isochrones, that starts at the high point of DC (0 km) and ends at LDC (at 40.7 km). However, we only show results starting at EDC which is 6.3 km downstream of the DC summit, as this is where there are traced isochrones to constrain the model. BELDC is located at 39.8 km along the flow line and its results are shown

in Sect. 3.5.





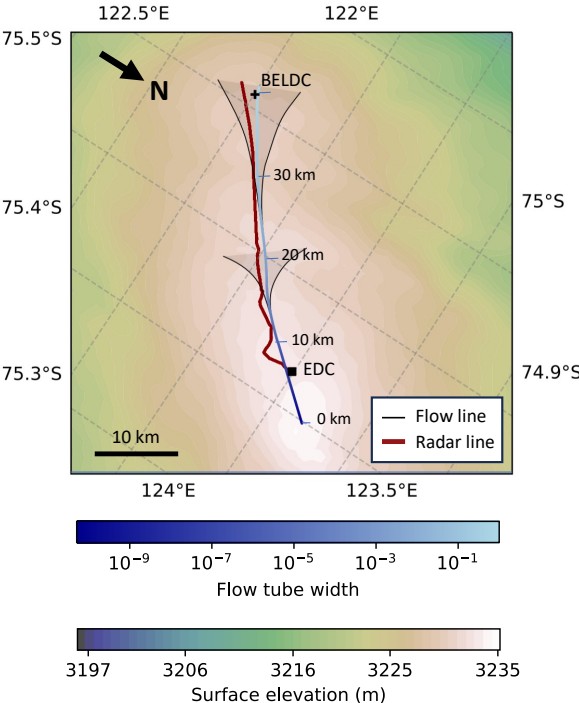

**Figure 2.** Map showing two flow tube section (shaded area spanned by black lines) from LDC to DC starting at two locations along the central flow line. We show two normalized flow tube sections along the same flow line for display clarity. The blue line indicates the central flow line, with shade corresponding to the overall flow tube width normalized to its widest point. Distance is marked along the flow line every 10 km and corresponds to distance on the x-axes of Figs. 3, 4, 5 and 6). Contours show surface elevation. BELDC is marked by a black cross and EDC is marked by a black square. The dark red line is the EDC–LDC radar transect from the LDC–VHF radar dataset in Chung et al. (2023a).

## 3.2 Ice particle trajectories

Figure 3 shows the ice particle trajectories originating from the surface as determined by the model. Through most of the upstream section of the flow line, including at EDC, ice particle trajectories reach the bedrock around 5 km downstream from where the snow was deposited on the surface.

Where the mechanical ice thickness from the model passes below the observed ice thickness, ice flow is still calculated. This results in some particle trajectories re-emerging from the observed bedrock. An example of this is the layer between the deepest trajectory which does not cross the observed bed and the top of the stagnant ice, over the LDC area beginning at distance > 32 km (blue, labelled "accreted ice" in Fig 3). Over LDC the thickness of this layer of potentially accreted ice is 120 m at 34 km, decreasing to ∼40 m thickness at the end of the flow line, near BELDC. See Sect. 4.1 for a full discussion of 180   this layer and its implications.





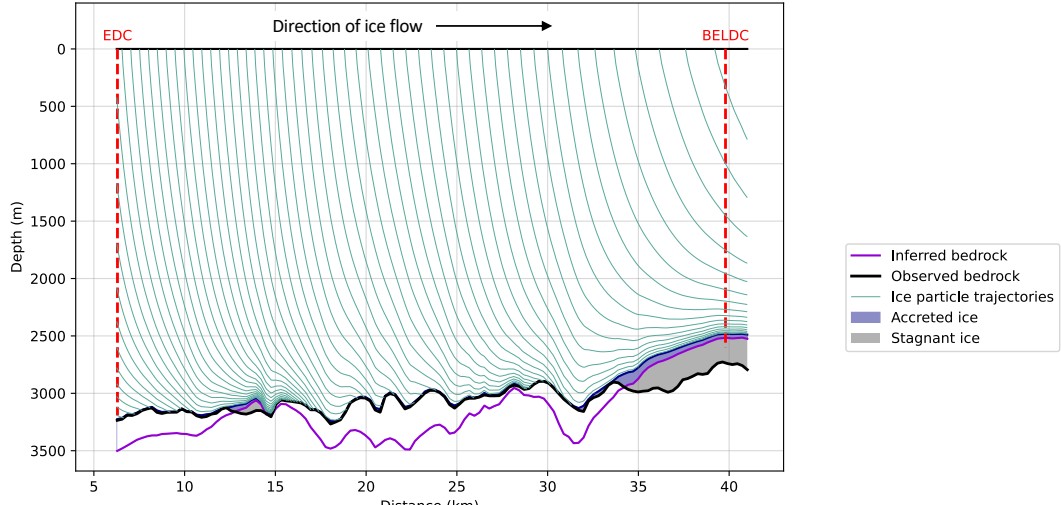

**Figure 3.** The thin green lines show ice particle trajectories along the DC–LDC flow line. The thick black line shows bedrock depth observed in the radar transect $H_{obs}$, the thick purple line is mechanical ice thickness $H_m$. Grey areas are modelled stagnant ice and the blue areas are possibly areas of accreted ice with trajectories originating near the bedrock. The locations of the EDC and BELDC ice core drill sites are marked by red dashed lines. The present day direction of horizontal ice flow is left to right, from DC to LDC.

### 3.3 Age of the ice

Figure 4 shows the age–depth relationship along the DC–LDC flow line. The modelled isochrones (in black) almost cover the observed isochrones which are in white behind, indicating a good fit. The associated age uncertainty is shown in Fig. S2.

In order to assess the validity of the model, we show the age misfit between modelled and observed isochrones (Figure 5). When comparing this to the age misfit of the 1D model in Chung et al. (2023a) applied to the DC–LDC flow line from this study (Figure S3), it is clear that the areas of over- and underestimation are similar.

### 3.4 Inferred parameters

Figure 6 shows the parameters inferred by the 2.5D model in this study compared to the results for the 1D model (Chung et al., 2023a) applied to the same flow line. The steady accumulation rate (Fig. 6) varies similarly for the 2.5D and 1D models
in the upstream region. However, there are large variations and uncertainties downstream for the 2.5D model. This is due to the fact that $\bar{a}$ at a given point affects the ice further downstream where there are no isochrone constraints and therefore residuals are unconstrained. The Lliboutry parameter $p$ has a strong peak at 28 km for the 2.5D model where there is a dip in the bedrock. Whereas the 1D model has a rounded peak as the averaging of the horizontal flow is taken into account. The Lliboutry parameter $p$ and the mechanical ice thickness $H_m$ show a downstream offset of around 1–2 km relative the 1D model.
The melt rate follows the same trend as the 1D model, however the variations are greater for the 2.5D model, again because the 1D inversion averages the inferred parameters along the horizontal travel of the ice. At EDC, the predicted melt rate is



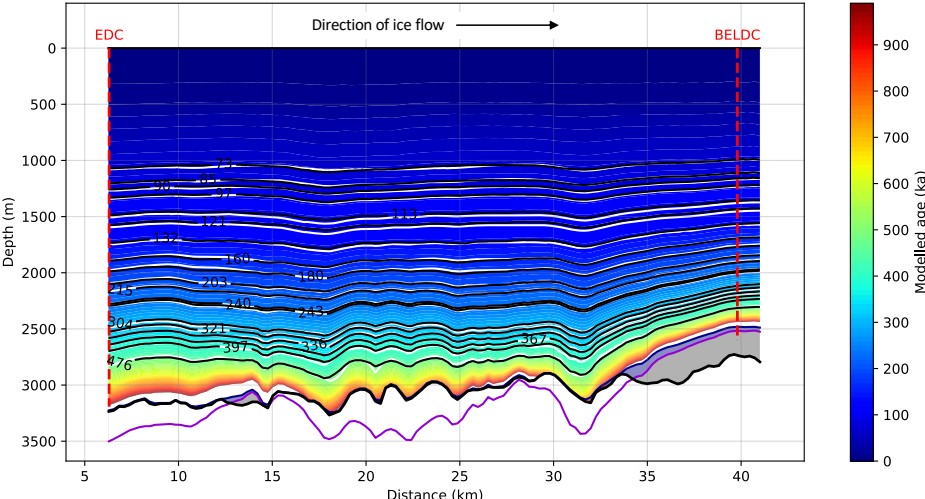

**Figure 4.** Age–depth profile from the 2.5D inversion model for the DC–LDC flow line. White lines show observed isochrones mapped from the radar transect onto the flow line and black lines show modelled isochrones. The thick black line shows bedrock depth observed in the radar transect, the thick purple line is mechanical ice thickness $H_m$. Grey areas are modelled stagnant ice and the blue areas are possibly areas of accreted ice with trajectories originating near the bedrock. The locations of the EDC and BELDC ice core drill sites are marked by red dashed lines. The present day direction of horizontal ice flow is left to right, from DC to LDC.

0.22 mm yr$^{-1}$ which agrees with previous studies (Passalacqua et al., 2017; Chung et al., 2023a). There is a small section of stagnant ice at a distance of 13 km (max. thickness 63 m) and a large layer of stagnant ice around 200 m thick over the LDC relief with a thin layer of accreted (see Sect. 3.2).

## 3.5 Results for the Beyond EPICA ice core (BELDC)

Here we look specifically at the results for the BELDC drill site, at a distance of 39.8 km along the flow line. The first novel result from this study is the origin of ice particles. Figure 7 shows that the deepest ice is formed from snow that fell around 15 km upstream. At a depth of 2452 m, the age density of ice becomes >20 kyr m$^{-1}$—the threshold for a measurable paleoclimatic signal (Fischer et al., 2013). At this depth, the ice is 1.12 Ma, significantly younger than the 1.5 Ma Beyond EPICA target. A high value for the inferred $p$ Lliboutry parameter of 14.0, results in an almost linear thinning function over the upper 2100 m of the ice sheet. For comparison, normalised vertical velocity measurements made with ApRES are shown in light blue in Fig. 7 with the dashed black line showing the best fit of these measurements from Chung et al. (2023a). The observed depth of the ice–bed interface $H_{obs}$ is 2735 m, which differs slightly from previous values of 2764 m (Lilien et al., 2021) and 2757 m (Chung et al., 2023a) likely because the bed surface is mapped onto the flow line so the closest location to BELDC is slightly different (Sect. 2.4). The mechanical ice sheet thickness provided by the inverse model is 2522±23 m. This combined with the observed depth of the ice–bed interface, results in a 214±23 m layer of stagnant ice, slightly larger than the value of 186 m from the 1D model with the same radar dataset (Chung et al., 2023a). There is a layer of accreted ice



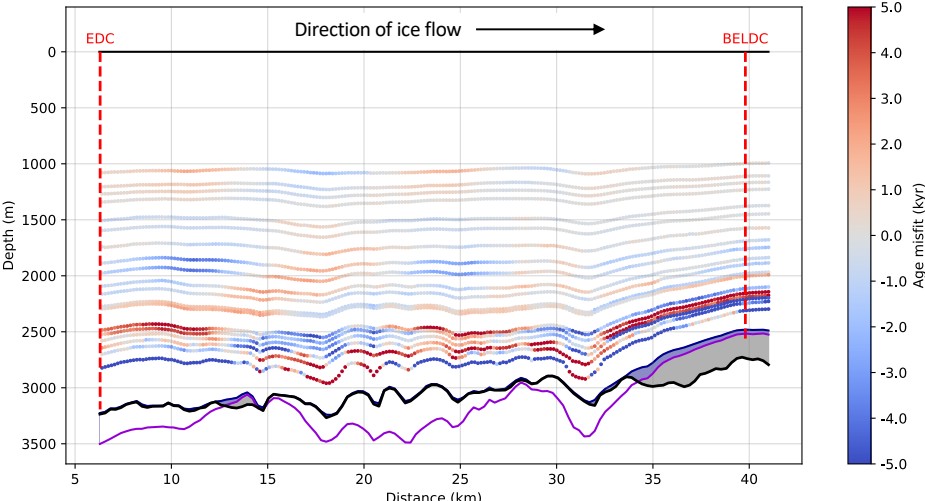

**Figure 5.** 2.5D model age misfit corresponds to the age of modelled isochrones minus the age of observed isochrones. The thick black line shows bedrock depth observed in the radar transect, the thick purple line is mechanical ice thickness $H_m$. Grey areas are modelled stagnant ice and the blue areas are possibly areas of accreted ice with trajectories originating near the bedrock. The locations of the EDC and BELDC ice core drill sites are marked by red dashed lines. The present day direction of horizontal ice flow is left to right, from DC to LDC.

37 m thick between the depth of the lowermost continuous trajectory (2485 m) and the top of the stagnant ice which is almost horizontally flowing according to the model (see Secs. 3.2 and 4.1). This combined with the stagnant ice leads to a total basal
layer thickness of 251 m.

## 4  Discussion

### 4.1  Model limitations

The 2.5D model calculates the average basal melt rate at EDC to be 0.22 mm yr$^{-1}$ (Sect. 3.4). However, it is likely that basal melting was slightly higher before 800 ka due to higher air temperatures (Lisiecki and Raymo, 2005; Passalacqua et al., 2017).
Higher basal melt rate could mean that the maximum age predicted by the model is overestimated in the melting regions, as it could have resulted in some of the oldest ice being lost to melting as likely occurred at EDC. However at LDC where there appears to be a thick basal unit, the melting process may not have affected the oldest ice which modelling suggests is not directly above the bedrock. Figure 6a shows a large uncertainty in the steady accumulation rate further from the dome, especially at distance >32 km (i.e. at LDC). This is because accumulation rates are not well constrained since there are no isochrones beyond
the LDC end of the radar transect. We provide a prior for accumulation ($a_{prior}$ in Eq. 2) in order to avoid too high variations in the downstream ill-constrained region. In Fig. 6b there are strong peaks in the inferred Lliboutry parameter $p$ for the 2.5D model where there are dips in the bedrock, indicating the effect of very local basal processes. We note that the exact location of





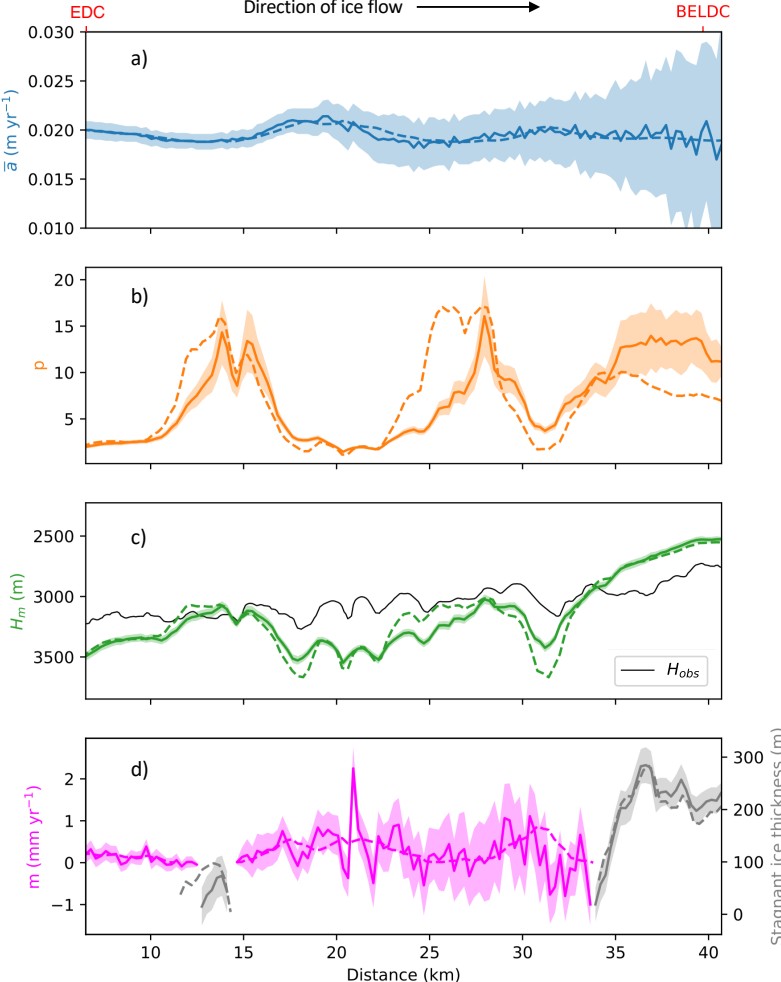

**Figure 6.** The parameters inferred by the 2.5D model with uncertainties shown by shaded areas. The 1D inferred parameters are shown by dashed lines. **(a)** average accumulation over the last 800 ka $\bar{a}$, **(b)** Lliboutry $p$ parameter, **(c)** mechanical ice thickness $H_m$ (green) and observed ice thickness from radar observations ($H_{obs}$, thin solid black line), **(d)** modelled basal melt rate (pink) and stagnant ice thickness (grey).

these dips are affected by the mapping of the isochrones from the radar transect onto the flow line (full discussion in Sect. 4.2). Looking at Figs. 6b and c, both $p$ and the mechanical ice thickness $H_m$ are offset downstream by around 1–2 km relative the 1D model. This can be explained when we consider that the two models may have found different solutions to account for the shape of the observed isochrones. As the thinning function is relatively linear up to the deepest isochrone, the models must therefore extrapolate to the bedrock. But since the observed bedrock is not used as a model constraint, there are two solutions; either $H_m$ is small and $p$ is large (leading to an almost linear thinning function); or $H_m$ is large and $p$ is small (leading to a nonlinear thinning function). Our two model approaches found these end-members of the solution space: the 2.5D model




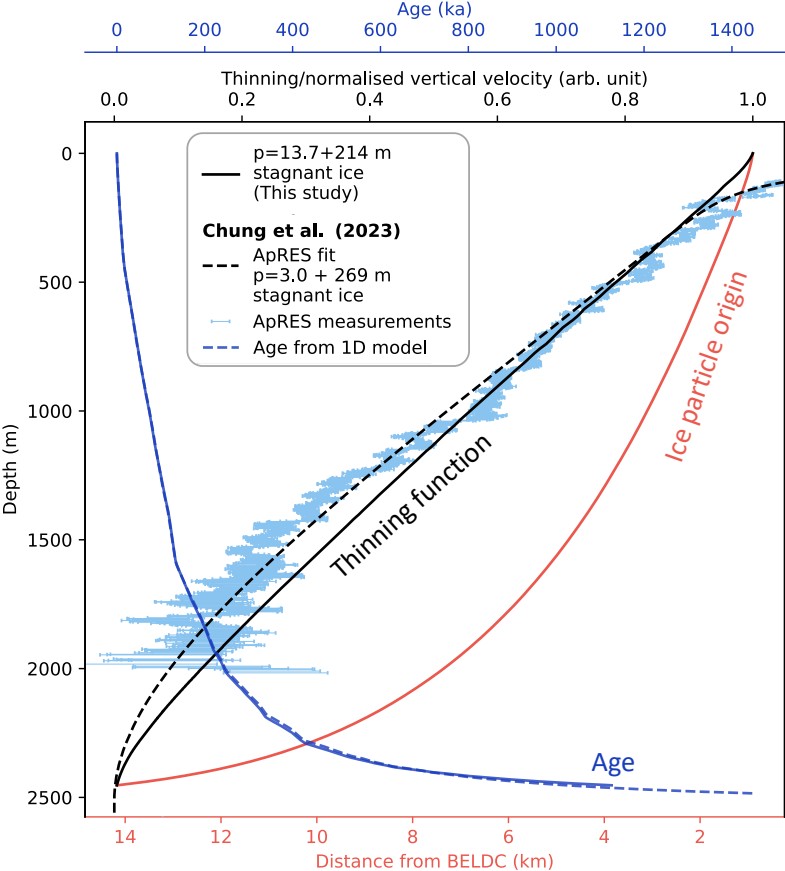

**Figure 7.** The dark blue solid line shows the age of ice changing with depth. The solid black line shows the normalised thinning function and the red line shows the horizontal distance origin of ice particles at BELDC from the 2.5D model. The dark blue dashed line is the age profile determined using the 1D model from (Chung et al., 2023a) applied to the DC–LDC flow line defined in this study. The light blue markers show ApRES of the normalised vertical velocity at LDC from (Chung et al., 2023a). The black dashed line shows the best fit for the profile, $p = 3.0$ and 269 m of stagnant ice.

found the first solution by taking into account the horizontal flow, the 1D model found the second solution, based on local calculations only. This suggests that horizontal flow is an important consideration, at least in these sections of the flow line. As the 2.5D model takes into account more physical processes, it has more information to narrow down the solution space, we therefore favour the 2.5D model result.

We ran the 1D model (Chung et al., 2023a), applied to the DC–LDC flow line and show the age misfit of the observed isochrones for this run (Figure S3). When comparing the misfits of the 2.5D and 1D models (see Figs. 5 and S3), the areas of over- and underestimation are very similar. There are a few possible explanations for the misfit which apply to both models; 1. The Lliboutry assumption may not be appropriate for this location; 2. Traced radar IRHs may not be accurate due to





the resolution of the radargram; 3. The subjective nature of tracing the IRHs or; 4. The position of the constraining radar observations with respect to the model flow line. The similarity of Figs. 5 and S3 suggests that the horizontal flow considered only by the 2.5D model cannot entirely reconcile the model isochrone age misfits.

The ice particle trajectories of snow which falls at EDC and further upstream indicate that it reaches the bed around 5 km downstream from where it was deposited. This is due to basal melting, which means that the vertical velocity, even at depth, is not negligible compared the horizontal velocity. In the last 10 km downstream and including at BELDC, influence from a bedrock dip and the stagnant ice result in larger horizontal movement of ice particles, meaning that ice comes from snow deposited around 15-20 km upstream. Due to ice particle trajectories re-emerging from the observed bedrock, the 2.5D model produces a layer of ice from 40-120 m thick at LDC (>34 km), on top of the stagnant ice. Downstream of the mountain at around 34 km, according to the model some accretion is required to provide an upward component of the flow trajectory. The model would seem to indicate that this layer is flowing horizontally or that there is a shearing zone over the stagnant ice, though this does not seem physically likely. It could be that as the model setup does not permit accretion at the observed bed, it shows accretion at the mechanical bed instead. This accretion could explain where the stagnant ice originates. Therefore, as it makes more physical sense, we discuss this "basal layer" as a whole (blue+grey in Figs. 3, 4 and 5). If this basal layer does indeed consist of some sort of accreted material, it could contain debris carried from the bedrock rise at 34 km over to LDC at around 200-250 m above the bedrock (Franke et al., 2023). The horizontal flow component could result in disturbed ice which may explain fragmented reflected layers observed in the LDC-VHF radargram (Lilien et al., 2021). These hypotheses above rely on the assumption of a steady geometry i.e. that the direction of ice flow between EDC and LDC and the locations of the dome and secondary dome have not changed. Although accumulation is low in this region of Antarctica, recent polarimetry measurements suggest that these assumptions are not necessarily appropriate (pers. comm. Carlos Martín). A transient model could include the movement of the dome over time, which could perhaps give an indication of flow in the lateral direction perpendicular to the radar transect.

Passalacqua et al. (2016) assessed the suitability of a 2.5D model along a divide. They found that in reality the flow tube does not have completely vertical walls due to ice flow changes in the past. This means that ice coming from areas adjacent to the flow line (i.e. not directly upstream) is not taken into consideration. This could be especially important at LDC where the mountainous bedrock relief may cause some bulging at the base of the flow tube as ice flows around the undulations in the bedrock. Previous studies have indicated the presence of strong anisotropy across DC (Ershadi et al., 2022). Thus, there may also be a complex coupling between the vertical velocity profile and anisotropy, which is not represented by the simple Lliboutry velocity profile used in this study (Eq. 1). A more complex model could take into account the effects of anisotropy, as Seddik et al. (2011) showed for the Dome Fuji region. Multiple anisotropic modelling efforts used idealised (Gillet-Chaulet et al., 2005; Gillet-Chaulet, 2006) or observed (Lilien et al., 2023) topography to examine how crystal fabric might affect flow around DC, finding notable changes to the flow field as a result of the crystal fabric. However, these more complex models are not easily differentiated to permit their use in arbitrary inverse problems and are currently too computationally costly to run in an inverse framework like that used here.



The 2.5D model in this study could be applied to other ice sheet flow lines where the location of the topographic dome has not moved significantly during the period of interest. Such locations include the flow line from Dome Fuji to the EPICA Dronning Maud Land ice core drill site (Steinhage et al., 2013), Dome B to Vostok (Siegert and Kwok, 2000) or even in Greenland from the GRIP to NorthGRIP and NEEM (Buchardt, 2009) or from the GRIP to EGRIP ice core drill sites (Gerber et al., 2021). These flow lines already have some traced radar IRHs, so the model could be applied without the need for new field campaigns.

## 4.2 Limitations of available observational data

In order to determine flow tube width, we use observed surface velocities from Vittuari et al. (2004) around DC and more recent observations over LDC (Sect. 2.3). As the horizontal flow is so low, continuing measurements over a longer time period would reduce uncertainties in the direction and magnitude of the surface velocity. The surface velocities were defined relative to DCRU (at Concordia station). However, we extrapolated the flow line to start at the highest point of the dome, 6.3 km upstream. Given that the velocities are comparable in magnitude to tectonic movements, we consider any uncertainty due to this extrapolation negligible.

The method used to map the radar IRHs onto the flow line (Sect. 2.4) introduces significant uncertainties, as at this resolution there are local physical features (such as bedrock dips or rises) that are included in the radargram but don't exist along the flow line and vice versa. The distance between the radar transect and flow line varies up to 1.9 km. There are clear deviations of the radar transect from the flow line at the EDC end where the station and different sectors had to be avoided. At the LDC end, the deviation is due to the fact that the BELDC site had not yet been selected. This survey was done as well as it could be given the restrictions. However, as a rule of thumb for flow models, it would be beneficial to conduct a radar survey following the flow line where possible. For this radar transect, there are no isochrones shallower than 1000 m. This was the result of a technical compromise so that the radar system could better image IRHs deeper in the ice sheet. Shallower isochrones are used to constrain accumulation rates and therefore it would be beneficial to include some. Such isochrones have been traced in other radar datasets (Chung et al., 2023a; Cavitte et al., 2021). However, given the different radar systems used and slightly different location of the transects, combining these observational data would lead to some isochrones having certain features where others from a different transect would not (Winter et al., 2017). Ideally, future surveys for similar purposes will tune settings to capture both shallow and deep isochrones. The deepest isochrone used reaches around 89% of the ice thickness, which is already a significant achievement. With a future chronology at BELDC, we could date deeper IRHs which were traced in the radargram over LDC but did not reach EDC for dating. The grid that the forward model uses (Figure S1) is very dense close to the dome. However, as there are no radar horizons upstream of EDC, this means that there are no age constraints where the grid nodes are most dense. To improve this, future radar surveys could extend from EDC to the DC summit.

## 4.3 Relevance and implications for BELDC

The maximum age of measurable ice at Beyond EPICA from the 2.5D model is 1.12 Ma at age density of 20 kyr m$^{-1}$. This is significantly lower than previous estimates using 1D models (Fischer et al., 2013; Parrenin et al., 2017; Chung et al., 2023a)



and is therefore a direct result of the consideration of horizontal flow in this study, combined with basal melt along the path that ice follows to reach the BELDC site. As ice flowing from the direction of DC encounters the mountainous bedrock relief at LDC, the ice sheet thickness decreases, effectively squeezing the layers and increasing thinning. This increases age density. Therefore the threshold of 20 kyr m$^{-1}$ is reached when the ice is younger. Whether the 1D or 2.5D model value is more reliable is open to debate, as either approach has its shortcomings. On one hand, the 2.5D model incorporates more physical processes.

However, the steady state assumption of constant flow direction or a stationary dome, used in this study, may not be appropriate in this area. For example, if the location of the dome was mostly around LDC during the past, this might mean that the 1D model is more appropriate for inferring the BELDC age scale.

From the comparison in Fig. 7 of the shapes of the thinning function from this study and the normalised vertical velocity from ApRES in Chung et al. (2023a), we can see that the 2.5D model has a much more linear shape due to the high value of $p$. This

could be due to the different depth ranges covered by the constraining observations — ApRES; 0-2000 m and radar isochrones; 1000-2300 m. In the upper part of the ice sheet, the thinning function is generally more linear, it is the lower part of the ice sheet that affects the shape of the thinning function the most. Therefore, given the the model's use of deeper constraints, it's results may be closer to reality. It is important to note that ApRES is locally informed in space and time while the model is integrated in space and time, therefore care should be taken with a direct comparison. The modelled normalised vertical velocity profile

would in fact be even more linear than the thinning function. This is because modelled thinning function accounts for higher ice thickness upstream which leads to more thinning.

In Sect. 4.1, we discussed the possible origin of the ~ 250 m thick stagnant/accreted basal layer. Each of the scenarios have notable implications for the deepest ice at BELDC: 1. Debris could be picked up by ice flowing over the bedrock rise at 34 km which would then be in the ice column at 200–250 m above the bedrock. This is much higher than previously suspected and

could provide the unique opportunity to obtain samples from the subglacial environment beneath the ice sheet. 2. Ice disturbed either due to the presence of debris or folding would considerably complicate continuous paleoclimatic signal preservation. However, discrete or partially continuous signals may still be retrieved. 3. If the layer is made entirely of accreted ice due to the freezing of subglacial water, any paleoclimatic signal we have been eliminated. However, its presence and studying its isotopy may inform attempts to reconstruct past ice sheet dynamics, such as change in overall ice sheet thickness. Any combination of

these scenarios is also possible, but little more can be deduced until ice samples are retrieved.

Another factor which should be taken into account when evaluating the current results is that numerical models using the Lliboutry assumption significantly overestimate the age in the deepest part of the ice sheet for both the Dome Fuji (DF) and EDC ice cores. Wang et al. (2023) showed that at DF, there is an inflection point around 400 m above the bed (Fig. 8b) where age from the chronology based on direct ice core measurements (Dome Fuji Ice Core Project Members, 2017; Oyabu et al.,

2022) deviates significantly from the exponential profile produced by a 1D pseudo-steady model using the Lliboutry vertical velocity profile. Similarly, Obase et al. (2023) found this issue when using their 1D transient model to compute the age and temperature evolution for the DF ice core. In this study (Fig. 8a), we show that the modelled age scale for EDC also deviates from the observation constrained chronology AICC2023 (Bouchet et al., 2023) at around 200 m above the bed, though the effect is not as severe as at DF. However, it is important to note that both previous ice cores (EDC and DF) were subject to





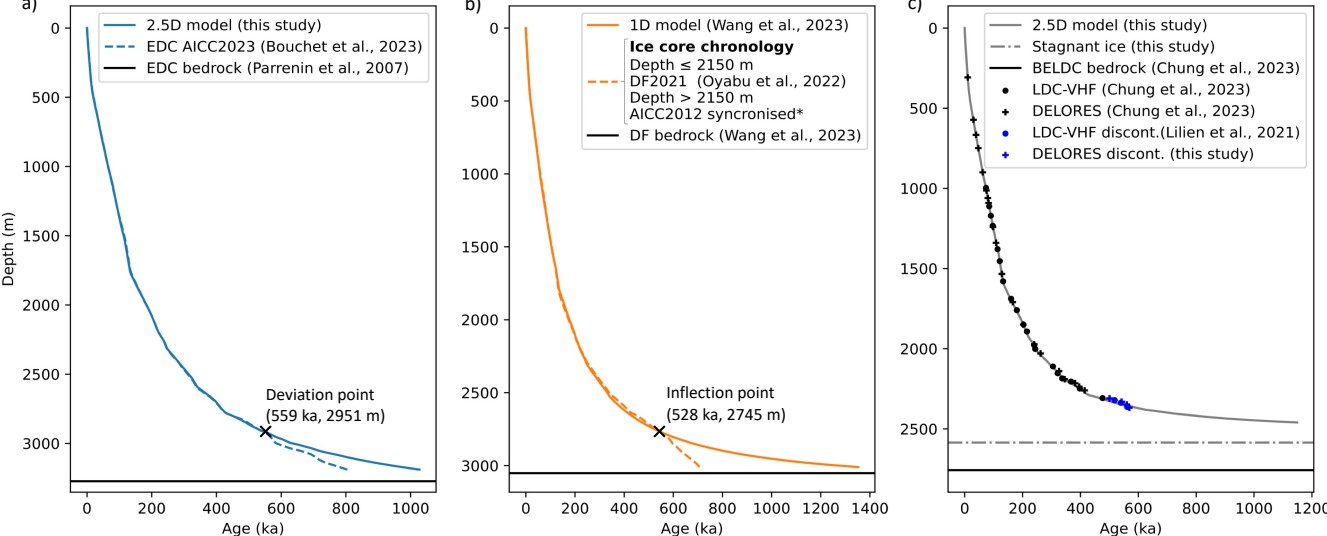

**Figure 8.** Modelled vs. observed ice core profiles. **a)** EDC age-scale from the 2.5D model presented in this study compared to the AICC2023 chronology (Bouchet et al., 2023). **b)** DF age-scale from a 1D model (Wang et al., 2023) compared to the combined ice core chronologies DF2021 (Oyabu et al., 2022) and a scale syncronised with AICC2012 proposed by *Dome Fuji Ice Core Project Members (2017). **c)** BELDC age-scale from the 2.5D model presented here. Black dots show isochrones used to the constrain the model in this study from the LDC-VHF dataset. Black crosses show DELORES isochrones (Chung et al., 2023a) for comparison. Blue dots and crosses show discontinuous isochrones traced in the respective datasets.

basal melting which may have contributed to decreased thinning near the base of the ice sheet. It is therefore difficult to judge how this will affect the ice at BELDC given the unknown nature of the 200–250 m thick basal layer (Chung et al., 2023a).

    Fig. 8c shows seven points from discontinuous isochrones traced from EDC to BELDC, three from the LDC-VHF dataset (previously published in Lilien et al., 2021) and four from the DELORES dataset (Chung et al., 2023a), deeper than those previously published. These discontinuous isochrones are only shown to indicate how the model compares at these depths
where no other possible constraints currently exist. However, they are not reliable enough to use in the main simulation. It seems that the three deepest discontinuous isochrones at BELDC may begin to show the downward trend of an inflection point in the age scale despite the fact that there is a basal layer between it and the bedrock. This could be a true inflection point as seen in the DF ice core (Fig. 8b) making the age significantly younger, or it may follow the step shape of the EDC ice core meaning that the difference from the model would be smaller. It is also possible that the inflection point does not occur due
to the basal layer, and the deepest discontinuous isochrones are simply displaying oscillations around the smooth Lliboutry thinning profile.



# 5 Conclusions

We presented a 2.5D pseudo-steady state inverse ice flow model applied to the flow line from DC to LDC. We found melting between EDC and the edge of LDC which is in agreement with previous studies. At LDC we find a 200–250 m thick basal layer which could be stagnant, accreted or disturbed ice, perhaps containing bedrock debris. Our model shows that the deepest ice at BELDC comes from around 15 km upstream with a maximum modelled age of 1.12 Ma and an age density of 20 kyr m$^{-1}$. The model could be applied to flow lines elsewhere in Antarctica if there are suitable radar data.

Our 2.5D model finds a different depth–age scale compared to 1D approaches, even in this area of slow horizontal ice flow. These differences arise due to melting and subsequent vertical strain along the flow line leading to BELDC. Such differences have to be taken into account when evaluating the reliability of age–depth relations based on 1D assumptions. Given the available constraints, a full 3D approach might not provide any greater accuracy than the 2.5D approach used here, unless a 3D observational radar data set for constraining the age–depth relation is used. However, this can only be answered definitively by a comparison of 2.5D and 3D approaches.

*Code availability.* Code for the 2.5D model will be made available on Zenodo

*Author contributions.* FP developed the forward age model and AC implemented the inverse method with input from FP. LV, MF and RM organised and conducted the collection of the surface velocity measurements and AZ was involved in processing the measurements. AC set up and ran all the simulations. AC prepared the first draft of the manuscript and all authors commented on and revised the manuscript.

*Competing interests.* The authors declare that they have no conflict of interest.

*Acknowledgements.* The authors would like to thank Carlos Martín for constructive discussions on the interpretation of modelled results and a radar observations. This publication was generated in the frame of Beyond EPICA. The project has received funding from the European Union's Horizon 2020 research and innovation programme under grant agreement No. 730258 (Oldest Ice) and No. 815384 (Oldest Ice Core). It is supported by national partners and funding agencies in Belgium, Denmark, France, Germany, Italy, Norway, Sweden, Switzerland, The Netherlands and the United Kingdom. Logistic support is mainly provided by ENEA and IPEV through the Concordia Station system. This publication was generated in the frame of DEEPICE project. The project has received funding from the European Union's Horizon 2020 research and innovation programme under the Marie Sklodowska-Curie grant agreement No 955750. The opinions expressed and arguments employed herein do not necessarily reflect the official views of the European Union funding agency or other national funding bodies. This is Beyond EPICA publication number XX. Marie Cavitte is a postdoctoral researcher of the FRS-FNRS.



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
