# Peer review of "Age, thinning and spatial origin of the Beyond EPICA ice from a 2.5D ice flow model"

_EGUsphere, 2024_

## Referee Report (RR1)

This manuscript combines a 2.5-D ice flow mode with IPR sounding data to study the basal ice dynamics at LDC, which has important implications for locating old ice and interpreting the paleoclimate record it preserves. This study addresses an important science question and will be of interest to a broad community. The methodology is robust and well described, and the manuscript includes clear and informative visual illustrations. The authors have responded thoroughly to the previous reviewer comments and revised the manuscript accordingly. I have only a few minor comments and suggestions that I believe could further improve the clarity and quality of the manuscript.

My primary concern relates to the treatment of input data uncertainties and their influence on the model outputs, particularly in regard to the interpretation of critical features such as the inferred basal accretion layer. While the authors provide a good overview of the sources and ranges of uncertainties (e.g., in divergence rates, radar line positioning, IRH depths, etc.), the manuscript would benefit from a more explicit evaluation of how these uncertainties propagate through the inversion process and affect key model outputs. For instance, the identification of an accretion or stagnant ice layer relies on a relatively small difference among $H_m$, $H_{obs}$, and traced particle trajectories. Without an assessment of their sensitivity to uncertainties in IRH positions or surface velocity measurements, it is difficult to fully trust the robustness of this interpretation. Maybe a minor underestimation in IRH depth or flow line misalignment could alter the model results significantly? A formal sensitivity analysis or at least a discussion of plausible error margins and their implications on interpreting basal processes would greatly strengthen the paper and provide readers with a clearer understanding of the reliability of the model's conclusions.

Another concern relates to the approach used to estimate the accreted ice layer thickness. The authors derive this by comparing the depth of the deepest traced ice particle trajectory with $H_m$. However, this method seems to assume that subglacial meltwater travels through the bedrock along similar paths as ice particles within the ice sheet, eventually refreezing along those trajectories. This assumption is problematic. The movement of meltwater within or beneath the bedrock is controlled by very different processes than ice flow, such as basal hydraulic potential gradients, bedrock porosity and permeability, and subglacial hydrological routing. It is not physically reasonable to assume that meltwater would follow the same spatial paths as deforming ice, especially over bedrock obstacles or variable basal conditions. I suggest the authors clarify this assumption.

Line 146-152: This paragraph, especially the second half, is hard to understand. The methodological details related to GNSS measurements and pole installation lack sufficient explanation, making it hard for the reader to grasp the technical reasoning. No references are provided to support the statements or the methodological details.

Line 246-247: It takes a long time for changes in surface temperature to propagate through the ice column and influence the basal thermal condition and basal melt rate.

Line 347-352: The observation that the 1D model appears to yield age estimates closer to preliminary field observations than the more physically comprehensive 2.5D model is intriguing. The authors suggest that this may be due to past migration of the ice divide/dome, but this hypothesis is not elaborated upon. It would significantly strengthen the manuscript if the authors could expand this discussion:

- Why would ice divide migration lead to better performance of a 1D model that assumes no horizontal flow?
- Could this discrepancy be interpreted as indirect evidence supporting past dome movement or reorganization of the ice flow?
- What would be the implications of such migration for site selection or future modeling approaches?

A deeper exploration of these possibilities would provide valuable insight into the glaciological dynamics of the Dome C region and could open a useful discussion about the limits and contextual applicability of 1D versus 2.5D modeling frameworks.

---

## Author Response (AR3)

Monday 2 June 2025

**Response to review 3 (Report #1)**

Reviewer comments
Author response
*Text added to manuscript*
Line numbers in manuscript submitted with this report

This paper presents a 2.5D inverse ice flow model to reconstruct age–depth profiles along the Dome C (DC) to Little Dome C (LDC) flow line in Antarctica, addressing limitations of conventional 1D models by incorporating horizontal ice flow dynamics. The model integrates radar-derived internal reflecting horizons (IRHs) and surface velocity data to constrain ice particle trajectories and basal conditions. Key findings include the identification of a 200–250 m thick basal layer at LDC, likely composed of stagnant, accreted, or disturbed ice, and an estimated maximum ice age of 1.12 million years (Ma) with an age resolution of 20 kyr m$^{-1}$. The model highlights the significance of horizontal advection, even in low-flow regions, and provides critical insights for the Beyond EPICA project's ice core drilling efforts.The 2.5D approach represents a significant improvement over traditional 1D models by accounting for lateral ice flow divergence and vertical strain , Although it's hard to say that this model is new. This enables more accurate age–depth predictions, particularly in regions with subtle horizontal dynamics.The identification of a thick basal layer at LDC offers a plausible explanation for ice sheet behavior, including potential stagnant ice or basal melt. This aligns with previous studies while introducing new hypotheses about ice accretion or debris entrainment.The framework is adaptable to other Antarctic regions with sufficient radar and velocity data, enhancing its utility for broader ice core projects. Thus, I believe that this paper is worthy of publication after addressing some of the concerns below.

We thank the reviewers for their time and appreciate their valuable comments and suggestions for improvement. We have replied to each comment below.

Major concerns

The model assumes steady-state flow conditions and static flow line geometry over 1.5 Ma, which may oversimplify glacial dynamics. Paleoclimatic shifts or ice sheet reorganization could alter flow patterns, introducing age uncertainties.Perhaps, a three-dimensional coupled thermomechanical ice flow model (such as Elmer/Ice) could more realistically simulate the ice flow conditions in the flux tube area.

As requested by reviewer 4, we have added further description on how potentially oversimplified glacial dynamics would affect the results.

New text (Lines 359-375)
*Whether the 1D or 2.5D model value is more reliable is open to debate, as both approaches have their advantages and shortcomings. On one hand, the 2.5D model incorporates more physical processes. However, the steady state assumption of constant flow direction or a stationary dome, used in this study, may not be suitable in this area. For example, if the location of the dome was mostly around LDC during the past, this might mean that the 1D model is more appropriate for inferring the BELDC age scale. This is because if the location of the dome and by extension the direction of flow from DC to LDC has reversed once or multiple times in the past, particle trajectories would be much more complex than the simple DC to LDC assumed by the 2.5D model. Therefore, the 1D model may average the effects of a reversal in horizontal flow direction, resulting in an age for the ice closer to reality than with the 2.5D*

*assumptions, despite the simpler model. Preliminary analysis of the BELDC ice core drilled to bedrock in the 24/25 season suggests the ice to be at least 1.2 Ma (Rannard, 2025), already older than the 2.5D model predicts. This could indicate that the assumptions of the 2.5D are not appropriate for the glaciological conditions in this area. If it is indeed shown that the 1D model provides a better age estimate at LDC on further analysis of the BELDC ice core, this would support the hypothesis that the dome has migrated during the period represented by the ice core. Though quantifying dome migration and flow direction changes would require other methods such as more complex 3D modelling or perhaps could be seen using ApRES measurements. Such a dome migration could be important to consider for future ice core site selection around DC. It would mean that very local glaciological features, e.g. the LDC mountainous bedrock relief, are likely to have a more significant effect on the age and continuity of paleoclimatic records preserved in the ice, than the conditions upstream as hypothesised in this study.*

Moreover, Radar IRHs were mapped to the flow line with deviations up to 1.9 km, potentially introducing local artifacts.

We have discussed this in lines 331-337, detailing as much as possible why the radar line deviates from the flow line. It is due to restrictions during the field work therefore this is the best data currently available. Improving the data would require a resurvey of the area, something that could be considered in future field seasons but will be logistically challenging

Shallow isochrones (<1000 m) were excluded due to radar resolution trade-offs, limiting constraints on accumulation rates.

Thank you for bringing up this point, we have added further discussion on this.
New text (Lines 261-264)
*The uncertainty shown in Fig. 6a is propagated from the uncertainty in radar isochrone depth. However, for accumulation, it is also important to note that as there are no isochrone constraints shallower than 1000 m (see Sect. 4.2). Accumulation uncertainty is larger due to lack of shallow isochrones, therefore we recommend caution with further interpretation of the inferred accumulation rate (Fig. 6a).*

Surface velocity uncertainties persist due to low ice flow rates (~mm/yr), comparable to tectonic movements, necessitating some detailed discussion.

The method for determining surface velocities takes into account the effect of tectonic movement as discussed in Lines 153-160. A more detailed description of this can be found in Vittuari et al. 2025 who used the same method for the data surrounding Concordia station.
Quantifying uncertainties in the LDC surface velocity measurements is currently difficult as there are only a few field season's of measurements. Repeating these measurements would allow us to better quantify uncertainties (suggested Line 328) but for now we must use the data we have.

Minor Concerns:

In Section 2. Methods, it would be preferable to provide formulas or definitions related to the steady accumulation rate, Lliboutry velocity profile parameter p, and mechanical ice thickness Hm. Otherwise, readers may find it challenging to comprehend the relationships between these parameters.

We have now defined steady accumulation

New text (Lines 106-109)
*The model performs a transformation from accumulation $a$ to steady accumulation $\bar{a}$ using a multiplicative temporal variation factor (Parrenin et al., 2017; Chung et al., 2023b) according to $\bar{a} = ar(t)$. $r(t)$ is the ratio of the accumulation at time $t$ inferred from the AICC2023 chronology (Bouchet et al., 2023) for the EPICA Dome C ice core (EDC, EPICA members, 2004) to it's temporally averaged value over the last 800 ka.*

$p$ is defined by the Lliboutry profile (Eq. 1, Line 112)

We feel that the definition of $H_m$ (Lines 123-131) is substantive and would not add more as we reference Chung et al. 2023 where there is a similar description as well as a figure and a more detailed discussion.

Line 85: The explanation of how variations in parameter p affect simulation results appears somewhat vague. It would be beneficial to either present a parameterized sensitivity analysis or cite relevant references to support this discussions

We now cite Parrenin et al. 2017 and Lliboutry 1979 who discuss the effect of change $p$ in more detail. (Line 116)

Section 4.1 "Model limitations" (starting at line 220) should consider addressing the influence of bedrock uplift or subsidence caused by tectonic movements on ice thickness evolution. Assuming the basal ice indeed has an age of 1.2 million years, over such prolonged timescales, the bedrock elevation would typically have undergone complex vertical changes. These tectonic adjustments would likely exert non-negligible impacts on ice dynamics and thinning processes.

Changes due to tectonic movements would not have an effect on the modelled flow, as this is a small area far from the ocean. The whole region would have undergone the same changes and therefore the flow pattern would not be affected.

It is the variations in ice thickness that are important as they can have an impact on the thinning function (Parrenin et al., 2007). The estimated max variations of ice thickness are ~150m, ~5% of current ice thickness (Parrenin et al. 2007, Ritz et al. 2001), so this is small, but future studies could aim to incorporate this physical process.

Other minor comments:
Line 8: While the study claims this is a new model, it appears to combine elements from Waddington et al. (2007) and F. Parrenin (2007; or 2011;2017). To substantiate its novelty, please provide specific technical details differentiating this model from previous methods, particularly in terms of equations, boundary conditions, or validation methods.

This model does use elements of previous publications. The new aspect is that it is an inverse model so optimisation of certain parameters can be performed. This is explained in the methods section in more detail. However we have added a sentence to the introduction to clarify this.

New text (Line 10)
*This 2.5D model uses a previously developed numerical scheme with the novelty being the inverse methods used to optimise multiple parameters by comparison to radar constraints.*

Line 12: The statement regarding "age density of 20 kyr m−1" requires clarification. Please specify whether this metric applies to the entire ice column or specifically to basal ice layers. Additionally, provide a brief technical definition of "age density" for readers unfamiliar with this particular terminology.

We have now reworded this to clarify the 20kyr/m threshold and removed the term "age density" from the abstract.

New text (Line 14)
*The threshold for ice useful for paleoclimatic reconstruction is 20 kyr m$^{-1}$ (20,000 annual layers per metre in the ice column). The oldest ice that meets this age resolution requirement is 1.12 Ma at BELDC according to the model.*

It is defined on first use in the main text and we have added more details on the 20k kyr/m threshold. (Lines 28-31)
*This is especially important for the deepest ice at the Beyond EPICA drill site (BELDC), as the age density (number of years per depth unit) is likely to be very high (Chung et al., 2023b), making extracting a paleoclimatic signal challenging. It is generally agreed that with current understanding and experimental techniques, ice with an age density of < 20 kyr m$^{-1}$ would be useful for paleoclimatic reconstruction (Fischer et al., 2013; Chung et al., 2023b).*

Line 14: The interpretation of ice composition ("stagnant ice, accreted ice or disturbed ice containing debris") appears speculative. We recommend either (a) presenting quantitative evidence from particle trajectory modeling to support these hypotheses, or (b) tempering the language to more explicitly acknowledge the interpretive nature of these conclusions.

This is our interpretation of the modelled results so we have edited the text to reflect this. However, until the ice core is analysed, all interpretations will remain speculative.
New text (Line 16)
*Looking at modelled ice particle trajectories, interpretations include that this layer could be composed of stagnant ice, disturbed ice, or even accreted ice, possibly containing debris.*

Lines 33-40: The basal unit description contains ambiguous terminology.
Whether the radar signature describes a stagnant ice layer with particular englacial characteristics? Or, the interpretation refers specifically to melt-refreeze processes as described by Bell et al. (2011),in which those radar imaging differ from/directly support previous interpretations of basal ice.

Regarding the terminology "basal unit", we follow earlier usage in the scientific literature, e.g. Goldberg et al. (2020) at a different place or Lilien et al. (2021) for our location. The term "basal unit" is generic and can only be interpreted in the context. First, for imaging with radar, it refers to the lowermost zone in a radargram above the bed reflection where radar imaging shows fragmented IRHs or incoherent backscatter or no return power at all. The implication being that this is a property of the ice and not the radar, which would be capable of imaging complete IRHs if there were any at this depth. Second, regarding modelling, as employed in our study or earlier (e.g. Lilien et al., 2021) the lowermost layer of the ice sheet can be considered stagnant, or at least not very actively contributing to ice flow according to the model assumptions. Third, preliminary results from the newly retrieved ice core at LDC indicate that the lowermost layer has different properties than the ice above, notably in dielectric values as well as ice water isotopes (pers. communication Beyond EPICA consortium).

All three observations are coincident in terms of the depth range. We therefore attribute these properties to the same lowermost layer in the ice sheet, termed basal unit. We rephrased the explanation of the basal unit in the text accordingly to avoid confusion.

The questions whether the ice of the basal unit is of meteoric origin but of larger age than the stratified ice above or was formed through basal accretion cannot be answered right now. It requires a full processing of the ice core (happening until mid 2026) as well as further drilling for absolute age dating on a replicate sample to be retrieved in the season 2025/26. We hope to be able to provide an answer by 2027.

Reworded text (Lines 42-47)

*At LDC, a "basal unit" has been observed in radar surveys presented by Cavitte (2017) and Lilien et al. (2021). This basal unit is a layer directly above the bed that seems to have different ice flow characteristics to the ice above. Chung et al. (2023b) showed that the basal unit seen in radar surveys is of comparable thickness to a stagnant ice layer predicted by a 1D inverse age–depth model. They then corroborated this with vertical velocity measurements made with an autonomous phase-sensitive radio-echo sounder (ApRES) which suggested that the ice layer above the bed is not flowing vertically (hence the name stagnant).*

Lines 101-102: It could be missing key references for the EDC ice core chronology. Please add citation.

We have referenced Bouchet et al. 2023 which is the reference for the AICC2023 ice core chronology and since this is the first mention of the EDC ice core we have added (EPICA members, 2004) (Line 108)

Lines 126-133: The ice flow velocity description lacks critical details.

We are not sure what the reviewer is referring to here. This paragraph (Lines 122-139) talks about the optimisation of accumulation $a$, mechanical ice thickness $H_m$ and the Lliboutry profile parameter $p$.

A detailed description of how these parameters link to the velocity field is provided in Parrenin et al. (2025).

Figure 1 should ideally include a schematic diagram indicating the specific location of LDC within the East Antarctic Ice Sheet. Unless readers are highly familiar with this region, it would be difficult for general audiences to easily discern its geographical position and spatial relationship relative to deep ice core sites such as Dome C and Vostok.

In Figure 1, we have now added an inset showing the location of the map area in Antarctica. The EPICA Dome C (EDC) ice core site is marked with a black square.

**References**

Bouchet, M., Landais, A., Grisart, A., Parrenin, F., Prié, F., Jacob, R., Fourré, E., Capron, E., Raynaud, D., Lipenkov, V. Y., Loutre, M. F., Extier, T., Svensson, A., Legrain, E., Martinerie, P., Leuenberger, M., Jiang, W., Ritterbusch, F., Lu, Z. T., & Yang, G. M. (2023). The Antarctic Ice Core Chronology 2023 (AICC2023) chronological framework and associated timescale for the European

Project for Ice Coring in Antarctica (EPICA) Dome C ice core. *Climate of the Past*, *19*(11), 2257–2286. https://doi.org/10.5194/cp-19-2257-2023

Chung, A., Parrenin, F., Steinhage, D., Mulvaney, R., Mart\'\in, C., Cavitte, M. G. P., Lilien, D. A., Helm, V., Taylor, D., Gogineni, P., Ritz, C., Frezzotti, M., O'Neill, C., Miller, H., Dahl-Jensen, D., & Eisen, O. (2023). Stagnant ice and age modelling in the Dome C region, Antarctica. *The Cryosphere*, *17*(8), 3461–3483. https://doi.org/10.5194/tc-17-3461-2023

EPICA members. (2004). Eight glacial cycles from an Antarctic ice core. *Nature*, *429*(6992), 623–628. https://doi.org/10.1038/nature02599

Goldberg, Madison L., Dustin M. Schroeder, Davide Castelletti, Elisa Mantelli, Neil Ross, and Martin J. Siegert. "Automated Detection and Characterization of Antarctic Basal Units Using Radar Sounding Data: Demonstration in Institute Ice Stream, West Antarctica." Annals of Glaciology 61, no. 81 (2020): 242–48. https://doi.org/10.1017/aog.2020.27

Lliboutry, L. (1979). A critical review of analytical approximate solutions for steady state velocities and temperatures in cold ice sheets. In *Z. Gletscherkde. Glazialgeol.* (Vol. 15, Issue 2, pp. 135–148).

Lilien, D. A., Steinhage, D., Taylor, D., Parrenin, F., Ritz, C., Mulvaney, R., Martín, C., Yan, J. B., O'Neill, C., Frezzotti, M., Miller, H., Gogineni, P., Dahl-Jensen, D., & Eisen, O. (2021). Brief communication: New radar constraints support presence of ice older than 1.5 Myr at Little Dome C. *Cryosphere*, *15*(4), 1881–1888. https://doi.org/10.5194/tc-15-1881-2021

Parrenin, F., & Hindmarsh, R. (2007). Influence of a non-uniform velocity field on isochrone geometry along a steady flowline of an ice sheet. *Journal of Glaciology*, *53*(183), 612–622. https://doi.org/10.3189/002214307784409298

Parrenin, F., Chung, A., & Martín, C. (2025). age_flow_line-1.0: a fast and accurate numerical age model for a pseudo-steady flow tube of an ice sheet. *EGUsphere Preprint Repository*. https://doi.org/10.5194/egusphere-2024-3411

Ritz, C., Rommelaere, V., & Dumas, C. (2001). Modeling the evolution of Antarctic ice sheet over the last 420,000 years: Implications for altitude changes in the Vostok region. *Journal of Geophysical Research Atmospheres*, *106*(D23), 31943–31964. https://doi.org/10.1029/2001JD900232

Monday 2 June 2025

**Response to review 4 (Report #2)**

Reviewer comments
Author response
*New text in manuscript*
Line numbers in manuscript submitted with this report

This manuscript combines a 2.5-D ice flow mode with IPR sounding data to study the basal ice dynamics at LDC, which has important implications for locating old ice and interpreting the paleoclimate record it preserves. This study addresses an important science question and will be of interest to a broad community. The methodology is robust and well described, and the manuscript includes clear and informative visual illustrations. The authors have responded thoroughly to the previous reviewer comments and revised the manuscript accordingly. I have only a few minor comments and suggestions that I believe could further improve the clarity and quality of the manuscript.

We thank the reviewers for their time and appreciate their valuable comments and suggestions for improvement. We have replied to each comment below.

My primary concern relates to the treatment of input data uncertainties and their influence on the model outputs, particularly in regard to the interpretation of critical features such as the inferred basal accretion layer. While the authors provide a good overview of the sources and ranges of uncertainties (e.g., in divergence rates, radar line positioning, IRH depths, etc.), the manuscript would benefit from a more explicit evaluation of how these uncertainties propagate through the inversion process and affect key model outputs. For instance, the identification of an accretion or stagnant ice layer relies on a relatively small difference among Hm, Hobs, and traced particle trajectories. Without an assessment of their sensitivity to uncertainties in IRH positions or surface velocity measurements, it is difficult to fully trust the robustness of this interpretation. Maybe a minor underestimation in IRH depth or flow line misalignment could alter the model results significantly? A formal sensitivity analysis or at least a discussion of plausible error margins and their implications on interpreting basal processes would greatly strengthen the paper and provide readers with a clearer understanding of the reliability of the model's conclusions.

As requested by reviewer 3, we have now expanded the discussion of uncertainties in modelled results. Uncertainties in IRH depth are propagated through the optimisation process and are shown with the optimised parameter values in Fig. 6. We also now discuss the effect of the lack of shallow isochrones on accumulation rate uncertainty.
New text (Lines 261-264)
*The uncertainty shown in Fig. 6a is propagated from the uncertainty in radar isochrone depth. However, for accumulation, it is also important to note that as there are no isochrone constraints shallower than 1000 m (see Sect. 4.2). Accumulation uncertainty is larger due to lack of shallow isochrones, therefore we recommend caution with further interpretation of the inferred accumulation rate (Fig. 6a).*

As surface velocity measurements at LDC, they only cover a few years of movement, so determining uncertainties from this is difficult. Measurement should be repeated over a longer time to be able to reasonably quantify uncertainties. (Line 327)

We can appreciate the reviewers suggestion for a sensitivity test and understand its value. Building on this study, we would like to test how results change if lateral flow divergence or non vertical flow tube walls are considered (Lines 303-308). In future work, where there is melting, we would like to use inverse methods to directly infer a melt rate rather indirectly using inferred Hm. This is more physical and would not result in this issue of particle trajectories passing below the observed bedrock depth and therefore perhaps make the flow behaviour of the accretion zone clearer. However, this would require changes to the model and represent a significant amount of work which is outwith the scope of this paper. We hope we will be able to provide these answers in the future.

Another concern relates to the approach used to estimate the accreted ice layer thickness. The authors derive this by comparing the depth of the deepest traced ice particle trajectory with Hm. However, this method seems to assume that subglacial meltwater travels through the bedrock along similar paths as ice particles within the ice sheet, eventually refreezing along those trajectories. This assumption is problematic. The movement of meltwater within or beneath the bedrock is controlled by very different processes than ice flow, such as basal hydraulic potential gradients, bedrock porosity and permeability, and subglacial hydrological routing. It is not physically reasonable to assume that meltwater would follow the same spatial paths as deforming ice, especially over bedrock obstacles or variable basal conditions. I suggest the authors clarify this assumption.

First, we would like to highlight that this is our interpretation of what **could** cause this modelled "accretion zone", it is one possible explanation. As requested by reviewer 3 we have made this clearer in the abstract.
New text (Line 16)
*Looking at modelled ice particle trajectories, interpretations include that this layer could be composed of stagnant ice, disturbed ice, or even accreted ice, possibly containing debris.*

However, with the accreted zone, we are not suggesting that melt water necessarily follows the ice particle trajectories as we agree the behaviour of liquid water would be very different to that of ice. We are suggesting that ice melts at certain points, then at some point further downstream, water from another source freezes onto the base of the ice sheet. Of course modelling the movement of water would require a different model.

We understand that the confusion may come from how we worded this section, so we have reworded it to make it clearer.
Reworded text (Line 209-211)
*Where the mechanical ice thickness from the model passes below the observed ice thickness, ice flow is still calculated. As a result, some ice originates from particle trajectories that begin beneath the observed bedrock. We have named the layer where this occurs "accreted ice" (blue layer in Fig 3).*

As discussed above for the previous point, in future work we plan to change how the basal state of the ice sheet is represented in the model to see how that affects the behaviour of the deepest ice.

As mentioned in our review 3 response, there is also evidence from radar observations of a basal layer that behaves differently to the ice above (Lilien et al. 2021 and Chung et al. 2023). Preliminary results from the newly retrieved ice core at LDC indicate that the lowermost layer has different properties than the ice above, notably in dielectric values as well as ice water isotopes (pers. communication Beyond EPICA consortium). The questions whether the ice of the basal unit is of meteoric origin but of larger age than the stratified ice above or was formed through basal accretion cannot be answered right now. It requires a full processing of the ice core (happening until mid 2026) as well as further drilling for absolute age dating on a replicate sample to be retrieved in the season 2025/26. We hope to be able to provide an answer by 2027.

Line 146-152: This paragraph, especially the second half, is hard to understand. The methodological details related to GNSS measurements and pole installation lack sufficient explanation, making it hard for the reader to grasp the technical reasoning. No references are provided to support the statements or the methodological details.

We have now reworded (old) lines 149-152.
Reworded text (Lines 154-158)
*Given the small magnitude of the expected ice velocities, a guarantee of high repeatability in centring the geodetic antennas was required during the initial and repeat measurements at each site. This was achieved by using aluminium poles 3m long and 12 cm diameter, installed at a minimum depth of 1 m in the snow, with a forced centring mount for the antennas on the top of the pole, which acted as precise three-dimensional reference points (see Vittuari et al. 2004 and 2025).*

Line 246-247: It takes a long time for changes in surface temperature to propagate through the ice column and influence the basal thermal condition and basal melt rate.

It can take a few 10s of kyr for surface temperature to propagate down to the bedrock, so the MPT temperature change (if any) certainly has already propagated.

Line 347-352: The observation that the 1D model appears to yield age estimates closer to preliminary field observations than the more physically comprehensive 2.5D model is intriguing. The authors suggest that this may be due to past migration of the ice divide/dome, but this hypothesis is not elaborated upon. It would significantly strengthen the manuscript if the authors could expand this discussion:

• Why would ice divide migration lead to better performance of a 1D model that assumes no horizontal flow?
• Could this discrepancy be interpreted as indirect evidence supporting past dome movement or reorganization of the ice flow?
• What would be the implications of such migration for site selection or future modeling approaches?

A deeper exploration of these possibilities would provide valuable insight into the glaciological dynamics of the Dome C region and could open a useful discussion about the limits and contextual applicability of 1D versus 2.5D modeling frameworks.

Thank you for these questions, we have now made significant additions to this discussion to answer them.

New text (Lines 359-375)
*Whether the 1D or 2.5D model value is more reliable is open to debate, as both approaches have their advantages and shortcomings. On one hand, the 2.5D model incorporates more physical processes. However, the steady state assumption of constant flow direction or a stationary dome, used in this study, may not be suitable in this area. For example, if the location of the dome was mostly around LDC during the past, this might mean that the 1D model is more appropriate for inferring the BELDC age scale. This is because if the location of the dome and by extension the direction of flow from DC to LDC has reversed once or multiple times in the past, particle trajectories would be much more complex than the simple DC to LDC assumed by the 2.5D model. Therefore, the 1D model may average the effects of a reversal in horizontal flow direction, resulting in an age for the ice closer to reality than with the 2.5D*

*assumptions, despite the simpler model. Preliminary analysis of the BELDC ice core drilled to bedrock in the 24/25 season suggests the ice to be at least 1.2 Ma (Rannard, 2025), already older than the 2.5D model predicts. This could indicate that the assumptions of the 2.5D are not appropriate for the glaciological conditions in this area. If it is indeed shown that the 1D model provides a better age estimate at LDC on further analysis of the BELDC ice core, this would support the hypothesis that the dome has migrated during the period represented by the ice core. Though quantifying dome migration and flow direction changes would require other methods such as more complex 3D modelling or perhaps could be seen using ApRES measurements. Such a dome migration could be important to consider for future ice core site selection around DC. It would mean that very local glaciological features, e.g. the LDC mountainous bedrock relief, are likely to have a more significant effect on the age and continuity of paleoclimatic records preserved in the ice, than the conditions upstream as hypothesised in this study.*

**References**

Chung, A., Parrenin, F., Steinhage, D., Mulvaney, R., Martín, C., Cavitte, M. G. P., Lilien, D. A., Helm, V., Taylor, D., Gogineni, P., Ritz, C., Frezzotti, M., O'Neill, C., Miller, H., Dahl-Jensen, D., & Eisen, O. (2023). Stagnant ice and age modelling in the Dome C region, Antarctica. *The Cryosphere*, *17*(8), 3461–3483. https://doi.org/10.5194/tc-17-3461-2023

Lilien, D. A., Steinhage, D., Taylor, D., Parrenin, F., Ritz, C., Mulvaney, R., Martín, C., Yan, J. B., O'Neill, C., Frezzotti, M., Miller, H., Gogineni, P., Dahl-Jensen, D., & Eisen, O. (2021). Brief communication: New radar constraints support presence of ice older than 1.5 Myr at Little Dome C. *Cryosphere*, *15*(4), 1881–1888. https://doi.org/10.5194/tc-15-1881-2021

Vittuari, L., Vincent, C., Frezzotti, M., Mancini, F., Gandolfi, S., Bitelli, G., & Capra, A. (2004). Space geodesy as a tool for measuring ice surface velocity in the Dome C region and along the ITASE traverse. *Annals of Glaciology*, *39*, 402–408. https://doi.org/10.3189/172756404781814627

Vittuari, L., Zanutta, A., Gandolfi, S., Martelli, L., Ritz, C., Urbini, S., & Frezzotti, M. (2025). Decadal Migration Of Dome C Inferred By Global Navigation Satellite System Measurements. *Journal of Glaciology*, 1–48. https://doi.org/10.1017/jog.2025.28

**Response to editor decision**

We thank the editor for their time and consideration and understand their choice to ask for major revisions given reviewer comments.

We address the critical comments from reviewer 1 by now referencing the article in preprint Parrenin et al. (2025) throughout and adding more descriptions to the methods section.

We have also modified some of the wording choice around the age of the Beyond EPICA ice as some preliminary results have been released to the press but a full analysis of the core is still yet to take place and be published.

**Response to review 1**

Reviewer comments
Author comments
*New text in manuscript* (line number)

We thank the reviewer for their comments and address their concerns with the lack of description of the forward model here.
A companion article detailing the full analytical method used in the forward model is now available in preprint Parrenin et al. (2025). We hope that it provides the technical details that the reviewer requested here.

We would like to highlight that this article is intended as an application of an existing model at Dome C and a discussion of its effects on the Beyond EPICA drill site. We felt that including both the analytical model development and fully discussing the application would be too heavy for a single paper, and these 2 aspects are relevant for different audiences. Therefore, in this article we focus on the application at Dome C.

This study present a so called 2.5D inverse model for simulating the age–depth relationships along a flow line from Dome C to Little Dome C in Antarctica. This is an important topic in glaciology, e.g., finding a location of an old ice core, which matchs the scope of The Cryosphere. However, I find it is very difficult for me to understand this manuscript, and at some places I feel not much senses to me.

The analytical development is heavily based on the work of Parrenin and Hindmarsh (2007). The idea to infer a mechanical ice thickness is based on Chung et al. 2023 as mentioned in section 2.1.

The main reason is that the authors do not even put some basic explanations of their model. I do not undertand why "a full description of the forward model will be available in a separate subsequent article" is possible if they do not make it clear in this paper? For example, Eqn 1 gives the vertical velocity profile, but what does the horizontal ice velocity (flux) look like? Both horizontal and vertical ice velocity are critical in determing the shapes of age-depth profiles.

We apologise for the cryptic statement here. At the time we submitted this paper, we were still writing the full description of the model. However, we found that the Cryosphere does not allow an

article to be referenced as "in prep", hence the statement. We have now referenced the preprint, Parrenin et al. (2025), throughout.

The horizontal flux is shown in Eq. 1 of Parrenin et al. (2025) and is the integral of the flow tube width, snow/ice density and horizontal velocity. Horizontal velocity depends on the normalised stream function (first defined by Parrenin and Hindmarsh, 2007). For the forward model, in this study, melting flux is zero, therefore the horizontal velocity simplifies to take the form of the derivative of ω.

The authors also do not explain basically the inverse model, e.g., what is p_prior and why they need to use a term "mechanical ice thickess" since they have already the observed ice thickness data - this do not make much sense not using the observed data!

The inverse model is very much related to the method applied in Chung et al. 2023. Indeed, this is where there is a full description of "mechanical ice thickness" and how it's comparison to observed ice thickness leads to the inferences of the basal condition of the ice. The use of measured rather than inferred/mechanical ice thickness would preclude the existence of a basal layer.
However, given the reviewers comments, we have now re-described these aspects of the model as this is a relatively new method which the reader may not be familiar with.

Text modified in Sec 2.2 (lines 117-126):
*The inverse model performs an optimisation to infer three parameters at each horizontal position, x: steady accumulation, ā (as defined in Sect. 2.1); Lliboutry thinning parameter, p (Eq. 1); and mechanical ice thickness, Hm. The mechanical ice thickness (Hm, first defined in Chung et al. (2023b)) is the best fit ice sheet thickness when considering the shape of the isochrones given no basal melt rate (Fig 1. from Chung et al. 2023). The difference between the mechanical ice thickness, Hm and the observed ice thickness Hobs is used to determine either a basal melt rate m or the thickness of a stagnant ice layer, as in Chung et al. (2023b). Where Hm < Hobs, there is a stagnant ice layer of thickness Hobs − Hm. The label "stagnant" is used as the vertical velocity profile is assumed to be zero at depths greater than Hm. Where Hm > Hobs, there is basal melting m which is calculated using the value of the ice flux Δq over the horizontal distance interval Δx at depth Hobs,*

*(Eq. 2),*

*where Y is the flow tube width.*

Actually, the age-depth modeling is not new. If the authors can dig a bit in past literatures, they can easily find some nice and important papers, like Greve et al. (2002) and Rybak and Huybrechts (2003). Based on ice flow models and the Eulerian or Lagrangian methods, we can determine the age-depth relationships in a more physical way. I do not see the authors even mention these previous work in the Introduction section.

As mentioned above, this article is intended as an application of the model near Dome C, therefore the introduction is more focused on what has been done previously at this location, not on age modelling in general. A more modelling oriented introduction can be found in the Parrenin et al. (2025) preprint, including reference to Rybak and Huybrechts (2003).

Sentence now added to introduction (line 29):
*There are many types of numerical schemes which can be used to model ice flow and the age-depth relationship in an ice sheet (Greve et al., 2002, Rybak and Huybrechts, 2003).*

I am not sure what the "forward model" is about if they do not use a physical ice flow model. If I am not convinced the velocity field is correct, it is also hard for me to believe the inversed age-depth profiles are correct either. The inverse model is then just an optimization approach to find the numbers that match the radar chronology record, but without much physics inside.

The forward model refers to the model presented in Parrenin et al. (2025) which uses the coordinate system from Parrenin and Hindmarsh 2007. Both of these papers give a full account of the physical ice flow model, therefore we do not go into so much detail in the article. In this study, we highlight only the adaptations to these previous works in order to make the inverse model.

We modify the introductory paragraph in Sec 2.1 - Forward model (lines 90-92) to make this clearer:
*The forward model is based on the analytical development of (Parrenin and Hindmarsh, 2007), with a numerical model presented in our companion paper Parrenin et al. (2025). As the numerical scheme is fully described in these articles, here we outline the benefits of this method and the slight changes required to make this suitable for an inverse model.*

We follow this with a more detailed description of the mechanics of the model without going into the mathematical detail already provided by Parrenin et al. (2025) and Parrenin and Hindmarsh, 2007. (lines 92-100):

*We use a pseudo-steady model which includes a steady-state geometry and velocity profile. The model determines the path of particles through the ice sheet by considering the total ice flux q through a flow tube with width Y as in Parrenin et al. (2025). In this study, the flow tube width is variable along the flow line in the x direction but constant in the vertical direction z.*

*We use the equation for ice particle trajectories presented in Parrenin and Hindmarsh (2007) and build a grid that follows these trajectories. This method has the advantage that ice particle trajectories pass exactly through grid nodes so no interpolation is required in the forward model. Given the increasing horizontal flow speed along the flow line, this means that the grid along the x axis is finer near the dome and coarser further downstream. In the z direction, the grid is coarse near the surface and becomes finer towards the bed where ice layers have thinned considerably (for more information see Supplementary material and Fig. S1).*

The writings also make me confused often. For example,
L13, "the 2.5D model predicts a basal layer 200–250 m thick at the base of the ice sheet", what is "basal layer 200-250 thick"?
Reworded to (line 13): *the thickness of a modelled basal layer is 200-250 m at the base of the ice sheet. This is a layer of ice above the bedrock which seems to have different flow behaviour to the ice flowing above.*

L68, "There is no direct thermal representation in the forward model as we use an inferred mechanical ice thickness to determine a basal melt rate", this sentence comes from no where, and has no references and no explanations.
This sentence relates to Parrenin and Hindmarsh (2007) (for the forward model) and Chung et al. 2023 (for mechanical ice thickness) which are referenced in the sentences before and after this one. As this is not clear as a stand alone sentence we have now added more description.

We modify the introductory methods (Sec 2, lines 70-88) to:

*We present a 2.5D ice flow model that uses inverse methods, constrained by radar observed isochrones, to fit poorly known parameters. The basic forward ice flow model is based on the analytical development presented in Parrenin and Hindmarsh (2007). Their numerical scheme was then developed into a forward model by Parrenin et al. (2025). This method is particularly efficient, as it performs a coordinate transformation to a system where particle trajectories are linear and therefore straightforward to calculate. The forward model does this by considering the ice flux due to vertical compression and horizontal ice flow. The main change from the numerical scheme presented in Parrenin et al. (2025), is that here, the forward model has no basal melting. The basal state of the ice sheet (including basal melting) is instead accounted for by the inverse model. The inverse model works by finding the best-fit value of physical ice flow parameters in the forward model, which lead to the current state of the ice sheet ie. the shape of the radar isochrones. The inverse approach is similar to that of a 1D ice flow model presented by Chung et al. (2023b). We use three inferred parameters—the steady-state (i.e., time-independent) accumulation rate $\bar{a}$, the Lliboutry velocity profile parameter $p$ and the mechanical ice thickness $Hm$. The inferred mechanical ice thickness is used to determine either a basal melt rate or the thickness of a stagnant ice layer as in Chung et al. (2023b).*

*The observation based constraints required for our 2.5D model are the flow tube width (forward model) and radar isochrones (inverse model). The width of the flow tube was determined using geodetic surface velocity measurements (Sect. 2.3). The inverse model is constrained by isochrones along a radar transect which approximately follows the flow line (Sect. 2.4). The forward model is run for different values of the three inferred parameters, resulting in modelled ages for observed isochrones. The inverse model optimises a cost function by selecting the best-fit parameters, which minimise the age misfit of the isochrones generated by the forward model to the observed isochrones.*

L80: what is the form of "horizontal flux shape function"?
*The horizontal shape function has the form of Eq. 38 of Parrenin and Hindmarsh (2007) and Eq. 1 of this article.*

L97: no definitions for p and Hw
These are defined in the text at:
Line 82: *Lliboutry velocity profile parameter $p$ and the mechanical ice thickness $Hm$*
Line 103: *The horizontal flux shape function is defined by the Lliboutry vertical velocity profile ($\omega$, Lliboutry, 1979), which depends on the $p$ parameter, (Eq. 1)*
Line 109: *changing the value of $p$, making the vertical velocity profile more or less linear*
Line 116: *Lliboutry thinning parameter, $p$ (Eq. 1); and mechanical ice thickness, $Hm$. The mechanical ice thickness ($Hm$, first defined in Chung et al., 2023b) is the best fit ice sheet thickness when considering the shape of the isochrones given no basal melt rate (Fig. 1 of Chung et al., 2023b). The difference between the mechanical ice thickness, $Hm$ and the observed ice thickness $Hobs$ is used to determine either a basal melt rate $m$ or the thickness of a stagnant ice layer, as in Chung et al. (2023b).*

L110: What is delta q and delta x?
Line 122: *the value of the ice flux $\Delta q$ over the horizontal distance interval $\Delta x$*

Table 1: what is the spatial locations for these 19 IRHs?
Added to Table 1 caption:
*These IRHs are traced in the radar transect shown by the red line in Fig. 2*

These kind of major and minor issues make me feel very difficult to read and understand this manuscript. Thus, I suggest the authors take another careful round of modifications and re-submit the manuscript after they add the necessary inputs.
References:
Ralf Greve, Yongqi Wang, and Bernd Mügge. Comparison of numerical schemes for the solution of the advective age equation in ice sheets. Annals of Glaciology, 35:487–494, 2002.
Oleg Rybak and Philippe Huybrechts. A comparison of eulerian and lagrangian methods for dating in numerical ice-sheet models. Annals of Glaciology, 37:150–158, 2003.

References

Chung, A., Parrenin, F., Steinhage, D., Mulvaney, R., Martín, C., Cavitte, M. G. P., Lilien, D. A., Helm, V., Taylor, D., Gogineni, P., Ritz, C., Frezzotti, M., O'Neill, C., Miller, H., Dahl-Jensen, D., & Eisen, O. (2023). Stagnant ice and age modelling in the Dome C region, Antarctica. *The Cryosphere,* 17(8), 3461–3483. https://doi.org/10.5194/tc-17-3461-2023

Parrenin, F., Chung, A., and Martín, C.: age_flow_line-1.0: a fast and accurate numerical age model for a pseudo-steady flow tube of an ice sheet, EGUsphere [preprint], https://doi.org/10.5194/egusphere-2024-3411, 2025.

Parrenin, F., & Hindmarsh, R. (2007). Influence of a non-uniform velocity field on isochrone geometry along a steady flowline of an ice sheet. *Journal of Glaciology,* 53(183), 612–622. https://doi.org/10.3189/002214307784409298

**Response to review 2**

We thank the reviewer for their comments and provide a response to their suggestions below.

Chung et al. use a 2.5D flowband model and dated internal layers to estimate the age of basal ice along a flowline from Dome C to the Beyond EPICA drill site of Little Dome C. The model finds the best fit spatial pattern of accumulation rate, velocity shape function, and mechanical ice thickness, which can then be translated to either stagnant ice thickness or basal melt rate. The primary result is that the age of basal ice (for Little Dome C this is just above the layer of stagnant ice) is younger than previously suggested, which has implications for the Beyond EPICA ice core.
The modeling is novel and well described. The manuscript is clearly written with a clear and well supported conclusion. There is acknowledgement of the unknown basal process which adds to the excitement of what Beyond EPICA will find when the drilling is completed this season (assuming good fortune). The discussion of ice cores having younger ages than the model predicts is interesting and useful component. The paper is ready for publication, but I hope the authors will consider the including the points below.

The one area that I suggest more of is a discussion of the age results with Lilien et al., 2021. While Lilien et al., 2021 is mentioned in multiple places, it is not clear how this age scale differs. Lilien et al. suggested that a 1.5Ma age, rather than a 1.1Ma age, is likely to be reached, but I think this is not so much a difference in the depth-age relationship as in the definition of "interpretable ice", with this paper using a value of 20 ka/m while Lilien et al. find 14 ka/m. It is a bit difficult for

people outside of Beyond EPICA to keep the differences straight, so providing discussions and summaries of the differences more clearly is quite helpful.

We agree that the discussion between past age scales and the one presented here need to be made clearer. In fact we compare not only to the work in Lilien et al. 2021 but also to that of Chung et al. 2023 which both use a 1D model rather than the 2.5D model presented in this work.
We have therefore added some clarifications (line 334-342):

*The maximum modelled age of measurable ice at Beyond EPICA from the 2.5D model is 1.12 Ma at an age density of 20 kyr m−1. This is significantly lower than previous estimates using 1D models (Fischer et al., 2013; Parrenin et al., 2017; Lilien et al., 2021; Chung et al., 2023b) and is a direct result of the consideration of horizontal flow in this study, combined with basal melt along the path that ice follows to reach the BELDC site. As ice flowing from the direction of DC encounters the mountainous bedrock relief at LDC, the ice sheet thickness decreases, effectively squeezing the layers and increasing thinning. This increases age density. Therefore the threshold of 20 kyr m−1 is reached when the ice is younger than the 1D modelling in Chung et al. 2023a. Lilien et al. 2021 applied a similar 1D model at BELDC. However, they used the threshold of 60 m above the mechanical ice depth to define the maxiumum age, as this was the basal layer thickness at EDC. Moreover, given the unknown nature of the stagnant ice layer, using this criterion at BELDC may not be appropriate.*

A few other minor comments:
- The discussion of ice fabric is appreciated and not including ice fabric in your model is understandable. However, it seems like the effects of fabric could be parameterized through constraints on the shape of the velocity, i.e. the p value. This is obviously future work, but I think considerable progress could be made without trying to model fabric evolution.

As the value of p is optimised by fitting to the isochrone observations, there is only a single value per vertical profile. Therefore, in this type of model we cannot separate the effects of ice fabric from other factors which change the value of p.

- The last sentence of the abstract is frustrating. Why end on such a sour note? Can you finish instead with a concluding note about the progress you have made?

We wanted to be clear about the limitations of the model but we have rearranged the abstract to end more positively.
Line 5: *We present a 2.5D inverse model that determines the age–depth profile along a flow line from Dome C (DC) to LDC that is assumed to be stable in time. This means that flow line features such as flow direction and dome location have not changed over the time period considered.*

Line 20: *Given that the age estimate from the 2.5D model is younger than previous estimates, this work shows the importance of considering the representation of the effects of horizontal flow when modelling the age profile.*